

# Technical note: Applying equilibration systems to continuous measurements of $p\mathrm{CO_2}$ in inland waters

T. K. Yoon[1,a], H. Jin[1], N.-H. Oh[2], J.-H. Park[1]

[1]Department of Environmental Science and Engineering, Ewha Womans University, Seoul 03760, Republic of Korea
[2]Graduate School of Environmental Studies, Seoul National University, Seoul 08826, Republic of Korea
[a]Current affiliation: Environmental Planning Institute, Seoul National University, Seoul 08826, Republic of Korea

*Correspondence to*: J. H. Park (jhp@ewha.ac.kr)

**Abstract.** High-frequency measurements of the partial pressure of $CO_2$ ($p\mathrm{CO_2}$) are crucial in elucidating spatiotemporal dynamics of $CO_2$ evasion from inland water systems and their role in the global carbon cycle. However, direct measurements of $p\mathrm{CO_2}$ are scarce, and the currently used measurement systems have not been compared for long-term deployment under various field conditions. A literature review was combined with laboratory and field tests to evaluate the application potential of three widely used automated equilibration systems, including spray- and marble-type equilibrators and a membrane-enclosed $CO_2$ sensor, to continuous long-term or underway $p\mathrm{CO_2}$ measurements in an urbanized river system in Korea. Both equilibrators had a shorter response time than the membrane-enclosed sensor, well capturing large spatial variations of $p\mathrm{CO_2}$ during a transect study along a highly urbanized river reach. The membrane-enclosed sensor based on passive equilibration provided comparable underway measurements for the same river reach until $p\mathrm{CO_2}$ approached the upper detection limit of the sensor. The membrane-enclosed sensor deployed in a eutrophic river site could detect large temporal variations of $p\mathrm{CO_2}$ over several weeks. To tackle biofouling on the membrane that can reduce sensor accuracy over time, we suggest that antifouling measures, such as copper-mesh screening, be used for long-term deployments with maintenance intervals ranging from several days to weeks. The overall results suggest that the equilibrators are better suited for relatively short underway measurements than long-term deployment, whereas the membrane-enclosed sensor can be used for both the underway and long-term continuous measurements if the sensor has a proper detection range and can be protected by a biofouling-resistant covering.

# 1 Introduction

Streams, rivers, and lakes *breathe* as the terrestrial biosphere does (Baldocchi 2008) and as do we. The evasion of carbon (c) resulting from the breathing in inland waters is influenced by a complex array of terrestrial and aquatic processes, including inputs of organic and inorganic C from both terrestrial and autochthonous sources, biogeochemical transformations in water, and gas transfer across the water-air interface (Aufdenkampe et al., 2011). Recent synthesis efforts have highlighted the importance of the evasion of $CO_2$ and $CH_4$ from inland waters in the global C cycle (Cole et al., 2007; Battin et al., 2009;



Bastviken et al., 2011; Butman and Raymond, 2011; Raymond et al., 2013; Borges et al., 2015). Rising interests in C evasion from the global inland water systems and their implications for climate change have resulted in an exponential increase in the annual number of scientific publications that are retrieved with search terms "$CO_2$" and "lake" or "river" on Scopus, from a mere 11 in 1980 to 414 in 2015.

The evasion of $CO_2$ can be determined either by directly measuring the transfer of $CO_2$ between water and air or by estimating the flux based on differences in the partial pressure of $CO_2$ ($pCO_2$) between water and air and gas transfer velocity. One of the most common direct flux measurements is using a closed chamber in which the $CO_2$ released from a fixed area of the water surface accumulates during a relatively short measurement period (Podgrajsek et al., 2014). The chamber-based flux measurements have faced many technical challenges including the difficulty of deploying the floating chamber stably
over often turbulent water surfaces, unlike deploying on rather firm land surfaces such as soils, plants, and deadwood (Hunt, 2003; Pumpanen et al., 2004; Yoon et al., 2014). Moreover, the floating chamber can disrupt environmental conditions inside and across the chamber that regulate the gas transfer velocity (e.g., wind speed, water turbulence, and temperature). For example, Vachon et al. (2010) reported on the risk of overestimating $CO_2$ flux by the floating chamber due to a disrupted natural turbulence inside the chamber. Eddy covariance flux measurements can be used as an alternative method for direct
flux measurement, but the use of this technique has been limited to a small number of aquatic systems (Huotari et al., 2011; Podgrajsek et al., 2014).

    The indirect approach based on $pCO_2$ has been used more widely than direct flux measurements; and this method's performance has been evaluated as reliable across a wide range of aquatic systems where the gas transfer velocity can be estimated with supplementary environmental data (Raymond et al., 2012). The water-air difference in $pCO_2$ is determined
from $CO_2$ measurements in water and air and then incorporated into an air-water gas transfer model (Liss and Slaster, 1974; Deacon, 1977; Wanninkhof, 1992; Raymond and Cole, 2001; Wanninkhof et al., 2009). The $pCO_2$ can also be estimated using temperature, pH, and alkalinity by applying the carbonate equilibrium model. Because calculated $pCO_2$ data can be easily obtained from existing water quality databases, it has been widely used in estimating $CO_2$ evasion from local to global inland water systems (e.g., Li et al., 2013; Lauerwald et al., 2013; Raymond et al., 2013). However, Abril et al. (2015) have
recently warned that calculated $pCO_2$ in acidic, organic-rich inland waters can be overestimated by 50 to 300 % relative to direct $pCO_2$ measurements, due to a combined effect of high leverage of organic acids to alkalinity and a limited buffering capacity of the carbon system in these waters.

    Although various $pCO_2$ measurement methods have long been used in a wide range of aquatic systems (Takahashi 1961; Keeling et al., 1965; Park et al., 1969: Smethie et al., 1985; Kling et al., 1992), technical advances have been stimulated by
ocean $pCO_2$ studies; these include developments of automated equilibration systems and standardized measurement procedures and data management as part of larger research projects such as the International Ocean Carbon Coordination Project (Feely et al., 1998; Dickson et al., 2007; Pierrot et al., 2009). Compared to relatively narrow $pCO_2$ ranges in the oceans (~100–700 µatm) (Valsala and Maksyutov, 2010), the $pCO_2$ in inland waters ranges from less than 100 to higher than 10,000 µatm (Abril et al., 2015). Moreover, freshwater $pCO_2$ varies across space and time to a much greater degree than in



the oceans, as a result of large variations in environmental conditions, complex biogeochemical processes involved in C transformations and $CO_2$ evasion, and anthropogenic disturbances (Cole et al., 2007). For instance, temporal dynamics of phytoplankton metabolism can result in large diurnal fluctuations of $pCO_2$ (Nimick et al., 2011). Turbulence-enhanced $CO_2$ evasion from rapidly flowing waters can result in steep downstream gradients of $pCO_2$ from the upstream sources (Dawson et al., 2001; Abril et al., 2014). Inputs of labile organic matter of anthropogenic sources can enhance the evasion of $CO_2$

from polluted waterways in urbanized watersheds (Frankignoulle et al., 1998; Zhai et al., 2005; Griffith and Raymond, 2011).

There have been successful efforts in developing methods for continuous $pCO_2$ monitoring to address large spatiotemporal variability in $pCO_2$ across a wide range of inland water systems (Frankignoulle et al., 2001; Johnson et al., 2010; Crawford et al., 2015). A few sensor-based studies have been successful in resolving large temporal variations of $pCO_2$ in a few inland water systems (Johnson et al., 2010; Huotari et al., 2013). These monitoring techniques have usually been tested in

headwater watersheds, but have not yet been applied to long-term monitoring of $pCO_2$ in larger river systems where human impacts, such as high loads of organic pollutants, are severe. While equilibrators have been deployed successfully for continuous underway $pCO_2$ measurements in large rivers and estuaries (Frankignoulle et al., 1998; Griffith and Raymond, 2011; Bianchi et al., 2013; Abril et al., 2014), these efforts have been focused on spatial variability of $pCO_2$ rather than integrating both spatial and temporal variations to provide accurate estimates of $CO_2$ evasion. In addition, individual systems

developed for continuous $pCO_2$ measurements have not been compared for the consistency of measurement accuracy during long-term deployments. This study aims (1) to review advantages and disadvantages of widely used $pCO_2$ equilibration methods and automated equilibration systems that can be used for continuous monitoring of highly variable $pCO_2$ across time and various inland water systems; (2) to compare the accuracy and maintenance requirements of three selected equilibration systems (a spray- and a marble-type equilibrators and a membrane-enclosed $CO_2$ sensor) for field applications

in a series of laboratory and field cross-validation tests; and (3) to provide recommendations for addressing technical problems and maintenance requirements that can hamper continuous long-term or underway measurements of $pCO_2$ in inland waters.

## 2 A short review on equilibration methods for continuous $pCO_2$ measurements in inland waters

Various equilibration-based approaches have been used to measure the $pCO_2$ in inland waters. Because gas analyzers, such

as chromatographs, determine the concentration of a gas in its gaseous phase, the gas dissolved in water needs to be equilibrated between the liquid and an artificially created "headspace"; the gas concentration in the headspace air is then analyzed using a gas analyzer (Swinnerton et al., 1962). The equilibration-based $pCO_2$ measurements usually follow three steps: (1) equilibrating the $pCO_2$ between the water sample and the air of a fixed volume; (2) measuring the gas concentration in an air sample from the headspace using a gas analyzer; and (3) additional calculations and corrections for

converting gas concentrations measured by the analyzer to the $pCO_2$ under specific conditions of temperature and barometric pressure. Although various equilibration methods can result from the combinations of equilibration procedures and gas



analysis methods, they can be grouped into three categories: headspace equilibration, equilibrators, and membrane-based equilibration (Table 1, Figure 1). These equilibration methods can also be classified based on the combinations of system operation (*manual* vs. *automatic* water sampling and air flow circulation) and equilibration mechanism (*active* vs. *passive* gas transfer to the detector in terms of external energy input). Automated systems employ gas equilibration devices that equilibrate a stream of flowing water with a stream of air that is circulated through a $CO_2$ detector in a closed loop (Frankignoulle et al., 2001; Santos et al., 2012). Our review and cross-validation tests focus on three automated equilibration systems: spray- and marble-type equilibrators and a membrane-enclosed sensor (Table 1). Gas analysis is usually conducted by a gas chromatograph (GC) or an infrared gas analyzer (IRGA) and can be combined with an isotope ratio mass spectrometer (IRMS) and a cavity enhanced absorption spectroscopy for additional analysis of $CH_4$ or stable C isotopes of $CO_2$ and $CH_4$ (Friedrichs et al., 2010; Gonzalez-Valencia et al., 2014).

### 2.1 Headspace equilibration

The headspace equilibration is usually accomplished by manually shaking a water sample that has been collected in a closed bottle or syringe and allowed to settle for a given time (Kling et al., 1992; Hope et al., 1995). Shaking facilitates the equilibration of $pCO_2$ between the water sample and headspace air. Either $CO_2$-free or ambient air is injected into the headspace to detect a change in the headspace concentration of $CO_2$ following equilibration. The equilibrated air is sampled using a gas syringe, and then the $CO_2$ concentration is analyzed by either a GC or IRGA. This method has been widely used as a standard method for direct $pCO_2$ measurements over the last decades. However, measurement errors associated with the manual equilibration, along with discrete samplings, limit the use of the manual headspace equilibration method in investigating large spatiotemporal variations of inland water $pCO_2$, as these require technically more advanced, automated equilibration systems.

### 2.2 Equilibrators

Various automated systems of $pCO_2$ equilibrators have been used for both discrete and continuous measurements of $pCO_2$ in inland waters and oceans (Takahashi, 1961; Keeling et al., 1965; Park et al., 1969; Feely et al., 1998; Frankignoule et al., 2001). Two most widely used equilibrators are the spray- and marble-type equilibrators (Table 1). The spray-type equilibrator has been used as a standard method in oceanic $pCO_2$ monitoring studies since its introduction in the 1960's (Takahashi, 1961; Keeling et al., 1965; Feely et al., 1998; Dickson et al., 2007; Pierrot et al., 2009). There are several commercially available versions of the product (e.g., GO8050, General Oceanics, USA). Various spray-type equilibrators have been increasingly used with diverse inland water systems (Raymond and Hopkinson, 2003; Zhai et al., 2005; Crawford et al., 2015; Joesoef et al., 2015). Marble-type equilibrators have been developed to address monitoring conditions specific to inland waters such as high organic matter loads and turbidity (Frankignoulle et al., 2001; Abril et al., 2006). The equilibrator basically automates the manual shaking used in the headspace equilibration method; using a spray, nozzle, or showerhead (as termed in different papers) and marbles increases the water-air interface for gas exchange without manual shaking. These



*automatic* and *active* systems integrate water sampling, equilibration, and gas analysis in a loop by using water or air pumps

that are powered by external sources (Table 1). A water-immersible pump has been preferred to deliver water from the source into the equilibrator at a fixed flow rate (1–3 L min$^{-1}$). When water is sprayed from a nozzle, the $p$CO$_2$ is equilibrated in droplets with the headspace air within a chamber of the equilibrator (Takahashi, 1961). In marble equilibrators, the pumped water flows through the surface of marbles stacked in an equilibrator chamber, allowing gas exchange between the flowing water and the headspace air (Frankignoulle et al., 2001; Abril et al., 2006). Marbles not only increase the air-water

interface, but also reduce the volume of headspace air, enhancing the speed of equilibration. The equilibrated air continuously circulates in a closed loop linking the equilibrator headspace to a gas analyzer, which is mostly an IRGA. Discrete air samples can also be collected with a syringe for analysis of CO$_2$ and other gases on a laboratory or on-board gas analyzer.

     Both the spray- and marble-type equilibrators can achieve continuous, high-frequency $p$CO$_2$ measurements only when

sufficient power is supplied to operate the pump and gas analyzer, without any failure or malfunction of the system components. In other words, power supply and maintenance can limit the application of equilibrator systems to continuous monitoring of $p$CO$_2$ in certain inland water systems. For instance, a long-term observation at a remote site or using a buoy is often challenged by maintaining power supply for a sustained period of time. The components of an equilibrator system involving water pumps, tubing, nozzles, and marbles can be clogged up with small particles and large debris when the

system is deployed for long-term monitoring in turbid or polluted waters with high loads of organic matter and suspended sediment (Santos et al., 2012). Moreover, large surface areas of marbles can be more vulnerable to biofouling than the spray-type equilibrator. Periodic maintenance, at intervals of days to weeks, might enhance long-term accuracy and stability of the equilibrator system, yet no systematic efforts have been made to reduce the technical problems associated with long-term deployment in harsh field conditions. Another maintenance issue is dehydrating the airflow before entering a gas analyzer;

desiccants (e.g., Drierite) need to be replaced periodically, necessitating frequent maintenance. Therefore, longer-term deployments of equilibrator systems, other than discrete spot sampling and short-term underway measurements, require a suite of maintenance measures that should be tested prior to long-term deployment at selected monitoring sites.

## 2.3 Membrane-enclosed sensors

     An *automatic*, *passive* CO$_2$ equilibration system can be established when a CO$_2$ sensor enclosed in a water-impermeable,

gas-permeable membrane is directly placed in water for continuous monitoring of $p$CO$_2$ (Table 1). Johnson et al. (2010) proposed a $p$CO$_2$ monitoring system consisting of a diffusion-type IRGA sensor (e.g., GMP220, Vaisala, Finland) enclosed in a polytetrafluoroethylene (PTFE) membrane, in which gas equilibration occurs between the inside (sensor) and outside (water) of the membrane and the CO$_2$ concentration in the equilibrated air inside the membrane is detected by the IRGA and can be logged by a connected data logger. There are a small number of commercially available membrane-enclosed sensor

systems (e.g., eosGP, Eosense Inc., Canada; Mini-Pro CO$_2$, Pro-Oceanus Systems Inc., Canada). This *automatic*, but *passive* equilibration system does not require the energy for equilibration and water and air pumping and therefore has the advantage



of lower energy demand than standard equilibrator systems. The relatively compact "all-in-one" system from equilibration to detection is also convenient for deployment in various inland waters. The low cost required for establishing the system allows for the expansion of the number of measurement locations in order to benefit from multiple or replicated observations.

This system has been successfully used for the long-term monitoring of $p\mathrm{CO_2}$ in headwater streams (Crawford et al., 2013; Peter et al., 2014; Leith et al., 2015), peatland open water pools (Pelletier et al., 2013), and mangrove waters (Troxler et al., 2015). In addition, this method can be deployed not only in surface waters but also in other media including sediments, soils, and dead wood (e.g., Johnson et al., 2010; Leith et al., 2015; Troxler et al., 2015). The wide range of applicability constitutes an important advantage of the membrane-enclosed sensor, which allows a cross-system comparison of C transfer across

various watershed compartments (Johnson et al., 2010).

Potential problems of the membrane-enclosed sensor system, including long response time and biofouling, have not yet been adequately investigated. The equilibration by passive diffusion usually requires longer times to reach equilibration ranging from several to dozens of minutes (Santos et al., 2012). Longer response times may hamper detecting large spatial or temporal variations of $p\mathrm{CO_2}$. It remains untested whether the membrane-enclosed sensor can be used for continuous

underway measurements of $p\mathrm{CO_2}$ in large rivers and estuaries. Biofouling on the membrane surface can result in either over- or underestimations of $p\mathrm{CO_2}$ during long-term deployment, although previous studies have not reported any significant effects of biofouling during long-term deployments of the sensor in headwater streams with relatively low ranges of $p\mathrm{CO_2}$ (Johnson et al., 2010; Crawford et al., 2013; Peter et al., 2014; Leith et al., 2015).

## 2.4 Other membrane-based and hybrid equilibration systems

Membrane contactors have been used as an alternative membrane-based equilibration method; they are also commercially available for industrial applications (e.g., Liqui-Cel®). The membrane contactors allow an *automatic* and *active* measurement of $p\mathrm{CO_2}$ when coupled with an automated system involving a $\mathrm{CO_2}$ anlyzer (Hales et al., 2004; Santos et al., 2012). A bundle of polypropylene membrane tubes provide sufficient contact area between water and air. Carignan (1998) designed a similar system with silicon tubes. The proposed systems operate in a similar way as marble- or spray-type equilibrators: water is

pumped through the membrane contactor where $\mathrm{CO_2}$ is equilibrated between water and air and transferred to a connected detector. Although membrane contactors have been used in various inland waters, including boreal (Teodoru et al., 2011) and tropical waters (Abril et al., 2015; Teodoru et al., 2015), potential problems of clogging and biofouling might limit the use of this method; this system has been tested only in relatively clean surface waters and groundwater with low concentrations of dissolved organic matter and suspended solids (Santos et al., 2012). An *automatic* and *passive* sensor-

based system, without being enclosed in a membrane, has been combined with a floating chamber to detect $\mathrm{CO_2}$ concentrations in the chamber air that is equilibrated with water (Bastviken et al., 2015). This coupling of the floating chamber and the NDIR sensor has an advantage of reducing biofouling over the surface of the sensor; however, the response time and long-term consistency of measurement accuracy have not been tested systematically.



Acidification of water samples before equilibration has been proposed to allow for determination of total dissolved inorganic carbon (DIC) (Å berg and Wallin, 2014). Acidification converts carbonate ($CO_3^{2-}$) and bicarbonate ($HCO_3^-$) to $CO_2$; next, the $pCO_2$ in the acidified water representing total DIC is determined. Å berg and Wallin (2014) modified the discrete headspace equilibration by adding the acidification step. A water sample is added into a vial, which is pre-filled with HCl and $N_2$ gas; the vial is then placed on a GC autosampler, without manual shaking or gas sampling. The $pCO_2$, excluding $CO_2$ originated from $CO_3^{2-}$ and $HCO_3^-$, is calculated by the carbonate equilibrium equation. As a result, the acidification headspace method has advantages of high performance and saving time at low temperatures. Bass et al. (2012) designed an *automatic* and *active* membrane contactor equilibrator, to which the automatic acidification system was added.

Aforementioned, monitoring systems can be coupled with other analyzers than IRGAs, including a cavity ring-down spectroscopy (CRDS; Friedrichs et al., 2010) and an off-axis integrated cavity output spectroscopy (OA-ICOS; Gonzalez-Valencia et al., 2014), to allow concurrent measurements of multiple gases and/or stable isotope ratios in $CO_2$, $CH_4$, and $N_2O$. Santos et al. (2012) proposed a coupled $pCO_2$-$^{222}$Rn monitoring system that can provide $^{222}$Rn data in addition to $pCO_2$. Although these novel systems usually employ expensive detectors, the development of low-cost systems has also been attempted. For example, Bastviken et al. (2015) presented an application case of affordable $CO_2$ sensor modules, which had been validated in the use of environmental studies (Yasuda et al., 2012). The use of low-cost detectors can allow for a replication of monitoring sites, which is essential for unveiling spatiotemporal variations of $pCO_2$, but this has not been attempted by most $pCO_2$ monitoring studies

## 2.5 Cross-validation studies

Different equilibration methods have been used separately under the assumption that they would provide identical $pCO_2$ measurement results based on the same principle of gas equilibration. This simplistic approach has not taken into consideration many complicated technical issues arising from difficult monitoring conditions and the maintenance of detector precision. In a rare, thorough comparative study, Abril et al. (2015) presented general agreements of $pCO_2$ measurements between the headspace equilibration and marble-type and contactor equilibrators over a wide range of $pCO_2$ (0 – 15,000 µatm) and other water chemical properties in various inland waters from temperate to tropical systems. An earlier comparison established an agreement between the marble-type equilibrator and headspace equilibration (Abril et al., 2006). On the other hand, Johnson et al. (2010) showed some moderate agreements of $pCO_2$ between the headspace equilibration and the membrane-enclosed sensor in four boreal inland waters; the membrane-enclosed sensor tended to exhibit higher $pCO_2$ values than the headspace equilibration, particularly over high concentration ranges exceeding 2000 µatm. To establish a system for coupled $^{222}$Rn and $pCO_2$ measurements in groundwater discharged into coastal waters, Santos et al. (2012) compared a spray-type, a marble-type, and three membrane contactor equilibrators (Liqui-Cel) and a passive polypropylene membrane (ACCUREL® PP, Membrana GmbH, Germany) system. Although all systems produced similar results of $pCO_2$ in the laboratory tests with a groundwater sample (~12,000 µatm), the response times of the tested equilibration devices differed markedly. The polypropylene membrane exhibited a very long response time (82 min) as





compared with shorter response times of the other devices ranging from 4 – 18 min. Empirical evidence supporting the agreement between the membrane-enclosed sensor and other equilibration techniques is insufficient. In addition, more systematic cross validations are required to evaluate the response time and biofouling of the membrane-enclosed sensor in
comparison with other equilibration systems.

## 3 Materials and methods

### 3.1 Equilibration systems

Spray-type and marble-type equilibrators and a membrane-enclosed sensor system were built following the most widely used designs available in the literature (Table 1 and references therein). Syringes were used for headspace equilibration following
widely used procedures (Kling et al., 1992; Hope et al., 1995). A polypropylene syringe (60 ml; HSW Norm-Ject Luer Lock Tip; Henke-Sass Wolf GmbH, Germany) was used to take a 30 ml water sample and then 30 ml of ambient air. The syringe was shaken manually for 2 min; then a subsample of the equilibrated air was collected in a 50 ml gas-tight syringe (Swastik Enterprise, Gujarat, India). The gas sample, as well as an additional 30 ml sample of ambient air, was directly injected into a GC (7890A, Agilent, USA) fitted with a Supelco Hayesep Q 12ft 1/8"column during laboratory tests or stored in a pre-
evacuated vial for later analysis during field tests. The $p$CO$_2$ was calculated from CO$_2$ concentrations of the equilibrated air and ambient air samples, water temperature, and barometric pressure, based on Henry's law (Hudson, 2004).

The spray-type equilibrator was built of a spray nozzle (GG 3/8 - SS 15, Spraying System Co., USA) in an acrylic tube (inner diameter: 40 mm; outer diameter: 48 mm; height: 200 mm) based on designs commonly used in previous studies (Keeling et al., 1965; Feely et al., 1998; Raymond and Hopkinson, 2003). For the marble-type equilibrator, glass marbles
(diameter: 10 mm) were filled in an acrylic tube (inner diameter: 40 mm; outer diameter: 48 mm; height: 300 mm) (Frankignoulle et al., 2001; Abril et al., 2006). Water was continuously pumped into both equilibrators with a bilge pump (Tsunami T800, Attwood Co., USA) at 2.5 L min$^{-1}$. A diaphragm pump was used to circulate the equilibrated air through an air filter and a desiccant (Drierite) column between the equilibrator chamber and an IRGA (LI820, Li-Cor, USA; GMP343 flow-through model, Vaisala, Finland).
A membrane-enclosed sensor system consisted of a CO$_2$ transmitter containing a sensor called CARBOCAP® (GMT222, Vaisala, Finland) and a data logger (CR10X or CR1000, Campbell Scientific Inc., USA), as described in detail by Johnson et al. (2010). The sensor probe was enclosed in PTFE membrane tubing (200-07, International Polymer Engineering, USA). The open end of the membrane tubing was sealed with a rubberizing compound (Plasti Dip, Plasti Dip International, USA). Three CO$_2$ sensors, a data logger, and two batteries (12 V 7 AH, Rocket, Korea; 12 V 100 AH, Atlasbx, Korea) in series
were installed in two portable, custom-made plastic containers.

The CO$_2$ analyzers and sensors were calibrated in the laboratory using CO$_2$ gases of known concentrations (0, 500, 500, and 10,000 ppm) immediately before each laboratory or field test. In the case of long-term deployment of the membrane-enclosed sensor over several months, the sensor was checked for measurement accuracy during maintenance breaks at



intervals of $1 - 3$ months and, if required, calibrated against the same set of standards. The outputs of the membrane-
enclosed sensor were corrected by temperature and barometric pressure (Johnson et al., 2010).

## 3.2 Laboratory tests

The correspondence of the $p\text{CO}_2$ equilibration systems was examined in preliminary laboratory tests. First, the accuracy of
the membrane-enclosed sensor was validated against the headspace equilibration measurements over the $p\text{CO}_2$ range from
370 to 6,300 µatm. The membrane-enclosed sensor was placed in a closed 2 L Erlenmeyer flask filled with deionized water.
A water pump was used to circulate water and gas through a rubber septum into the flask. For each of the target $p\text{CO}_2$ values,
a given volume of concentrated $\text{CO}_2$ gas (99.99 %) was injected into the flask. The water pump was operated over 20 min to
bring $p\text{CO}_2$ into a constant level. When the reading of the membrane-enclosed sensor stabilized, 50 ml of water was
collected from the flask for a headspace equilibration measurement by replacing it with 50-ml of $\text{N}_2$ gas.

In the second test, the response and measurement accuracy of the three equilibration systems were compared using tap
water that was continuously filled into a 6 L container. Because equilibrators require continuous water inflow, they were
provided with tap water through the 6 L container that was exposed to the ambient air to maintain a constant $p\text{CO}_2$. The
membrane-enclosed sensor was placed in the same container so that all three equilibration systems were exposed to the same
level of $p\text{CO}_2$. All $p\text{CO}_2$ equilibration systems were within a narrow $p\text{CO}_2$ range during the test ($2,008 - 2,210$ µatm).

## 3.3 Field tests of the spray-type equilibrator and the membrane-enclosed sensor

From May $4^{th}$ to $6^{th}$, 2015 accuracy and response times were compared between the spray-type equilibrator and membrane-
enclosed sensor at 12 sites, spanning from a forested headwater stream (38°15' N, 128°7' E, 582 m.a.s.l.) through stream and
river locations blocked by dams or weirs to the estuarine reach along the Metropolitan Seoul (37°41' N, 126°39' E, 1 m.a.s.l.)
of the Han River in South Korea. The river system is intensively dammed with more than ten large dams and several old and
newly built weirs. The predominant flow condition of each site was distinguished between standing and flowing waters
based on the distance from the closest dam or weir up- or downstream and the specific flow conditions during the field study.
The marble-type equilibrator was not used due to logistical reasons and instead the spray-type equilibrator was applied,
under the assumption that both types would exhibit similar results based on the laboratory test results. The level of $p\text{CO}_2$ and
other environmental conditions at the 12 selected sites were heterogeneous enough to conduct the cross-validation test. For
example, dissolved oxygen (DO), pH, and dissolved organic carbon (DOC) ranged $3.5 - 11.6$ mg $\text{L}^{-1}$, $6.8 - 9.0$, and $1.0 - 5.1$
mg C $\text{L}^{-1}$, respectively.

The water $p\text{CO}_2$ at a 20-cm depth was determined using simultaneously the headspace equilibration, membrane-enclosed
sensor, and spray-type equilibrator systems. The membrane-enclosed sensor was directly placed into the surface water at 20
cm below the surface. A peristaltic pump (Masterflex E/S portable sampler, Cole-Parmer Instrument Co., USA) was used to
collect water into a sample bottle for the headspace equilibration measurement, while a bilge pump was used for the spray-
type equilibrator. Except for the different type of IRGA (GMP343 flow-through model, Vaisala, Finland) that was coupled





with the spray-type equilibrator to enhance the portability in the field, the same measurement procedures and instrumental set-up as in the laboratory tests were used for three equilibration systems. Headspace equilibration and membrane-enclosed sensor measurements were performed at 6 of the 12 sites every month, from July 2014 to July 2015 in order to obtain additional data including the response time under various field conditions. The response time was determined as the full time

($t_{100}$) or 95% of the full time ($t_{95}$) it took to a final $p$CO$_2$ level that represents $p$CO$_2$ values exhibiting less than 1 % of coefficient of variation (CV) for 2 min.

### 3.4 Continuous underway measurements of $p$CO$_2$

To test the applicability of the three equilibration systems to continuous underway measurements of $p$CO$_2$ along a river reach receiving varying loads of organic matter via tributaries, a boat expedition was undertaken along the upper estuary of the

Han River (37°31' N, 127°1' E, 7 m.a.s.l.) on May 11, 2015. The selected river reach is influenced strongly by the inflow from several urban streams, including Tan Stream and Joongnang Stream draining from the Seoul metropolitan area. Large spatial variations in $p$CO$_2$ and other water quality components along the confluence with the urban stream were expected to create ideal conditions for a cross validation of the three equilibration systems. After prior tests of boat speed effect on the measurement accuracy of the three equilibration systems, the speed was maintained at ~10 km h$^{-1}$ over the distance of ~10

km, which falls in the usual boat speed range used for other continuous underway measurements (Abril et al., 2014; Crawford et al., 2015).

The water $p$CO$_2$ at 20 cm below the surface was continuously measured at intervals of 1 or 5 sec by the three equilibration systems. One membrane-enclosed sensor, together with a bilge pump that delivered collected water into the spray- and marble-type equilibrators and another membrane-enclosed sensor were attached to a pole and placed 20 cm below the water

surface on one side of the boat. A portable multi-parameter meter (Orion 5-Star Portable, Thermo Scientific, USA) was used to simultaneously measure water temperature, pH, electrical conductivity, and dissolved oxygen in the continuously collected water on board, while air temperature and barometric pressure were recorded in a micro-logger (Watchdog 1650 Micro Station, Spectrum Technologies Inc., USA). The two membrane-enclosed sensors (one placed in the river water and the other one immersed in the pumped water on board) had an upper detection limit at 10,000 and 7,000 µatm, respectively.

Along the 10 km transect, four headspace equilibration measurements were conducted to compare the measurement accuracy of the three equilibration systems.

### 3.5 Continuous long-term measurements of $p$CO$_2$

Several laboratory and field tests were conducted to examine the application potential of the three equilibration systems to continuous long-term monitoring in a river reach of the Han River receiving large loads of organic matter from up- and in-

stream sources and the tributaries polluted with wastewater treatment effluents and urban runoff. With preliminary tests showing that it was easier to maintain power supply and air flow dehydration than with the marble-type equilibrator, the spray-type equilibrator was selected for use. Long-term measurement stability of the spray-type equilibrator was tested



against that of the membrane-enclosed sensor in a series of unmanned field deployments. However, large power consumption by pumping and the gradual clogging of the nozzle resulted in repeated failures of the system. The resulting

$pCO_2$ data exhibited abnormal patterns $2 - 3$ d following the initiation of the monitoring. Therefore, long-term performance and antifouling measures of the membrane-enclosed sensor became the focus.

As part of a long-term monitoring project, a membrane-enclosed sensor ("membrane sensor") was deployed at a 20 cm depth below the surface along an uninhabited island on the downstream reach of the Han River near the city center of Seoul (37°32' N, 126°55' E, 5 m.a.s.l.) over the course of one year starting July 2014. To examine the effectiveness of the copper

mesh screening for reducing biofouling on the membrane surface, another membrane-enclosed sensor covered with copper mesh ("membrane+Cu sensor") was deployed at the same site for 43 d from May 31[st] to July 12[th], 2015. The membrane and membrane+Cu sensors were attached to a buoy, ~3 m off a dock constructed along the island shore. Two automobile batteries (12 V 100 AH in series) powered the sensors and placed on the island together with the $CO_2$ transmitter and a data logger. The power generally lasted two weeks, maintaining supply to two in-stream sensors and an additional sensor used for

the concurrent measurement of air $pCO_2$ 1 m above from the water surface. During routine biweekly maintenance visits, the membrane surface was cleaned with soft cloth and brush and then rinsed with deionized water; lastly the copper-mesh screen was replaced. In addition to $CO_2$, pH, DO, water temperature, conductivity, and turbidity were monitored using a multi-parameter water quality sonde (YSI 6820 V2, YSI Inc., USA). All continuous data was logged at 10-min intervals. During the field test, extraordinary algal blooms occurred as a combined result of a severe drought, warm temperatures, and high

loads of nutrients discharged from water treatment facilities and polluted tributaries draining the Seoul metropolitan area; chlorophyll-a concentration increased from 21.1 mg m$^{-3}$ on June 2[nd] to 46.7 mg m$^{-3}$ on July 2[nd] (Water Information System of Korea; http://water.nier.go.kr).

### 3.5 Data analysis

The agreement of $pCO_2$ measurements among equilibration systems was evaluated by linear regression analysis and by

examining CV values across the monitoring sites. Differences in the response time ($t_{100}$) were compared between equilibration systems by $t$-test. The relationships between the response time and $\Delta pCO_2$ (defined as the difference between the initial and the stabilized final $pCO_2$ during deployment) were established for each of the flowing and standing water types. For the continuous long-term measurements, the relative difference of $pCO_2$ was calculated from the natural log-transformed ratio between values of the "membrane sensor" and the "membrane+Cu" sensor. The pH-$pCO_2$ relationships

were described by LOESS (locally weighted scatterplot smoothing; Cleveland and Devlin, 1998) to examine the viability of $pCO_2$ measurements arising from biofouling in time series data. All statistical analyses, including descriptive statistics, t-test, regression analyses, and LOESS, were conducted on R (R Development Core Team, 2011).




## 4 Results and Discussion

### 4.1 Cross validation of system performance

A series of laboratory tests established good agreements in the measurement accuracy of the three compared equilibration systems, as exemplified by the correspondence between the membrane-enclosed sensor and headspace equilibration measurements over a large range of $pCO_2$ (Figure 2a) and among the three systems at a given $pCO_2$ (Figure 2b). The CV of the measurements of the three equilibration systems was 2.4 %. The test of response time distinguished the fast response of the spray- and marble-type equilibrators (~4 min) from the slow response of the membrane-enclosed sensor (~15 min)

(Figure 2b).

Short-term continuous measurements of $pCO_2$ for 30 – 60 min at 12 sites also showed general agreements between the two equilibration systems and the headspace equilibration measurements over the $pCO_2$ range from 152 to 10,340 µatm (Figure 3). The measurement results of the three systems showed strongly positive pairwise correlations ($R^2 > 0.99$). The CV values were smaller than 5 % at all sites except site 3 and 8 (a stream and a river channel blocked by a weir, respectively), in which

$pCO_2$ values were even lower than the atmospheric $pCO_2$. The good agreements between the tested methods are consistent with other studies that have demonstrated the accuracy of the equilibrators (Frankignoulle et al., 2001; Santos et al., 2012; Abril et al., 2015) or the membrane-enclosed sensor (Johnson et al., 2010), although these previous comparisons were conducted separately for each equilibration system.

Similarly to the laboratory test, the spray-type equilibrator had shorter response times than the membrane-enclosed sensor

(t-test: $P < 0.001$; Figure 4). The spray-type equilibrator usually reached the level of $pCO_2$ equilibration within a few minutes (Figure 4a), whereas the membrane-enclosed sensor required a longer time to reach the same $pCO_2$ level (Figure 4b). Mean $t_{95\%}$ and $t_{100\%}$ for the spray-type equilibrator was 1 min 31 s and 2 min 36 s, respectively, with no difference between standing and flowing waters. The response time of the spray-type equilibrator falls within the usual range of response times reported for the spray- (8 min; Santos et al., 2012) and marble-type equilibrators (2 – 3 min; Frankignoulle et al., 2001; Abril

et al., 2014). However, the mean $t_{95\%}$ and $t_{100\%}$ for the membrane-enclosed sensor were 14 min 49 s and 18 min 18 s for standing waters and 6 min 58 s and 9 min 54 s for flowing waters, respectively. This result suggests that while the response time is mainly controlled by the difference in $pCO_2$ between air and water, different degrees of turbulence in different water flow conditions can significantly affect the gas diffusion velocity that can be described by the diffusion coefficient of Fick's laws. The longer response time of the membrane-enclosed sensor can be explained by the passive equilibration without any

physical process to facilitate equilibration underwater (Santos et al., 2012). In other words, the surface area of air-water interface over the membrane-enclosed sensor is much more limited compared to the spray-enhanced air-water gas exchange. Since water turbulence can enhance the equilibration efficiency of the membrane-enclosed sensor, the deployment time at flowing waters can be shortened as compared with the longer time required for the deployment in standing waters.

For both equilibration systems, the response time increased logarithmically with $\Delta pCO_2$ (Figure 4). Response time

increased with increasing $\Delta pCO_2$ (i.e., high water $pCO_2$), with steeper increases observed for the membrane-enclosed sensor,





particularly in flowing waters. The slope of the relationship was in the order of the membrane-enclosed sensor in standing water (10.4) > the membrane-enclosed sensor in flowing water (2.8) > the spray-type equilibrator (1.4). Water pumping required for the equilibrators might explain no clear difference in the response time of the spray-type equilibrator. Only small portions of the variations of response time observed for the membrane-enclosed sensor were accounted for by $\Delta p CO_2$

($R^2 = 0.29$ for standing water and 0.20 for flowing water), suggesting that water flow and other in-stream processes might also affect the response time.

The results suggest that the deployment time of the membrane-enclosed sensor for short-term (< 1 h) deployments as part of multi-site discrete monitoring should be carefully determined based on water flow conditions and expected $p CO_2$ levels. Since only a sufficient time of underwater deployment can ensure accurate measurements of $p CO_2$ by the membrane-

enclosed sensor, we suggest a minimum deployment time of 10 min for flowing waters and 20 min for standing waters. When the long response time poses an obstacle to multiple discrete measurements covering a wide range of locations within a limited time, the equilibrators could be a quicker alternative with the same level of measurement accuracy.

### 4.2 Continuous underway measurements

Except for the river sections that showed drastic changes in $p CO_2$, continuous underway measurements of the two

equilibrators and two membrane enclosed sensors were generally in good agreement with each other – within 10 % CV. (Figure 5). The two equilibrators produced almost the same results across the monitored reach. This comparison corroborates the accuracy of previous underway $p CO_2$ measurements that have tested the performance of each of the equilibrator types separately against headspace equilibration measurements (Frankignoulle et al., 2001; Griffith and Raymond, 2011; Abril et al., 2014). The $p CO_2$ measurements of the membrane-enclosed sensors generally corresponded well to those of the

equilibrator and headspace equilibration measurements. However, measurements deviated substantially (around 12:00) along the river segments where the inflow from a highly polluted tributary enriched in $p CO_2$ elevated the $p CO_2$ of the main stem above the upper detection limits of the two different sensors (Figure 5). In contrast to the long response times observed for discrete measurements at the 12 sites (Figure 4), the membrane-enclosed sensors exhibited reasonably fast responses to large $p CO_2$ fluctuations across most of river sections; although it failed to respond to rapid $p CO_2$ increases from the

relatively low value at the confluence (11:57) to the concentration peak (12:25) due to the limited detection range. Increased turbulence arising from the boat movement might have enhanced equilibration of the membrane-enclosed sensor. In addition, there was little difference in $p CO_2$ values measured by the in-stream sensor and another sensor immersed in the pumped water on board. The relatively high flow of the water pump (3 L min$^{-1}$) might have generated sufficient mixing for rapid equilibration.

The test results suggest that both the spray- and marble-type equilibrators can be used for underway measurements along waterways with $p CO_2$ greatly varying in space. However, it remains unresolved how long the measurement accuracy can be maintained during a long cruise along high-$CO_2$ waterways without maintaining the replaceable items including nozzles, marbles, and desiccants. Previous studies have used $CO_2$ sensors only for continuous monitoring at a few discrete sites



(Johnson et al., 2010; Huotari et al., 2013; Peter et al., 2014; Leith et al., 2015) rarely examining spatial changes in response
time. Our transect results demonstrate that the membrane-enclosed sensor could also provide reliable continuous underway
measurements in the inland water systems showing large spatial variations of $p$CO$_2$ as the monitored river reach. A proper
calibration of the sensor for the high range of $p$CO$_2$ should precede the deployment of the sensor in high-CO$_2$ waters. Based
on the in-stream and on-board sensors producing almost the same measurement results, we suggest that on-board
measurements with pumped water can be used as a safer method for concurrent measurements of $p$CO$_2$ and other water
quality components as a way to avoid damage by unknown underwater obstacles such as large floating debris.

### 4.3 Continuous long-term measurements

The $p$CO$_2$ measurements by the membrane-enclosed sensors with and without the copper mesh screen started to diverge
substantially 3 − 5 d following the biweekly maintenance (Figure 6). Sensor $p$CO$_2$ measurements without the copper mesh
screen ("membrane sensor") exhibited larger diurnal fluctuations than those taken with the other sensor protected with the
copper mesh screen ("membrane+Cu sensor") during later phases of the biweekly monitoring intervals. When daily averages
were compared to reduce diurnal fluctuations, the $p$CO$_2$ measurements by the "membrane sensor" were higher than those of
the copper mesh-wrapped sensor; the differences increased with increasing time from the maintenance day. The duration
during which relative differences of day-averaged $p$CO$_2$ between the two sensors remained within 10 % was 5, 2, and 7 d
since the routine maintenance on the 153th, 169th, and 182th day of the year, respectively.

The relationships between pH and $p$CO$_2$ were used to examine the increasing biofouling effects with progressing time
following the maintenance day (Figure 7). The pH-$p$CO$_2$ relationships for the "membrane sensor" shifted upward as time
progressed from the maintenance day, whereas those for the "membrane+Cu sensor" remained consistent over time (Figure
7a). If additional CO$_2$ molecules were produced from biofouling over the "membrane sensor", this could disturb the usual
pH-CO$_2$ relationship that can be explained by the carbonate equilibrium (Nimick et al., 2011). In addition, the relationships
between the daily CVs of pH and $p$CO$_2$ were stronger for the "membrane+Cu sensor" ($R^2$ = 0.91) than the "membrane sensor"
($R^2$ = 0.51) (Figure 7b). The consistent pH-$p$CO$_2$ relationships observed for the "membrane+Cu sensor" indicate the
reliability of measured $p$CO$_2$ values, but the method validation would require concomitant $p$CO$_2$ measurements using other
equilibration methods under different conditions with varying $p$CO$_2$ values.

The test results suggest that the membrane-enclosed sensor may be vulnerable to biofouling in polluted waters similar to
the studied site, which can amplify diurnal fluctuations of $p$CO$_2$. Long-term deployments of any $p$CO$_2$ equilibration systems
cannot evade the problem of biofouling. Repeated maintenance visits at short intervals of 3 − 5 d may be required for a long-
term deployment of the sensor without antifouling measures in an inland water site with high levels and large diurnal
fluctuations of $p$CO$_2$. Day-averaged $p$CO$_2$ values may be used as representative $p$CO$_2$ levels within a week from the
maintenance day, but the uncertainty level cannot be determined without concomitant measurements using other discrete or
continuous measurements that are not significantly influenced by biofouling. We recommend that the copper-mesh screen be
used to minimize biofouling effects as a cost- and energy-efficient measure. Antifouling techniques can be classified into



various categories, including mechanical (e.g., wiper, brush, water jet, and ultrasonic sound) and a biocide (e.g., copper, chlorine, and UV) approaches (Delauney et al., 2010). Currently, wiping and copper-based materials are commonly applied to various water quality probes. For instance, YSI Inc. supplies antifouling kits for water quality sondes, including wipers, copper mesh screens, copper-alloy guards, and copper tapes, given that these practices have been evaluated as effective in various inland and marine environments (YSI Incorporated, 2010). As compared with other biocides, relatively low toxicity of copper ensures effective application in aquatic environmental monitoring (Manov et al., 2004). Other mechanical antifouling techniques (e.g., brushing and wiping) may be applied to the membrane-enclosed sensor system; nevertheless, the copper-mesh screen might better fit into long-term $p\mathrm{CO_2}$ monitoring programs which require easy deployment, minimal maintenance, and low energy demand. Biofouling might be a negligible problem in oligotrophic waters. For instance, we deployed bulk sensors at a forest headstream for one week and a reservoir for two weeks in July 2015 during the same season as the antifouling test was conducted. Following the 1–2 weeks of deployment, the membrane surfaces did not exhibit any visible sign of biofouling (unpublished work).

Stable power supply is another important factor for successful, long-term continuous observation. A membrane-enclosed sensor consumes approximately 30 times less power than a single bilge pump for equilibrators. A set of automobile batteries generally lasted 2 weeks, maintaining a power supply to the three membrane-enclosed sensors. Using an analog timer or relay system, the power of the membrane-enclosed sensor can be turned on and off at a pre-set interval. We estimate that two automobile batteries ($2\times12$ V 100 AH in series) can power one membrane-enclosed sensor for up to 3 or 6 months, assuming 30-min measurement operation for 2 or 4 h intervals. Measuring $p\mathrm{CO_2}$ at intervals of 2 or 4 h can provide enough data for daily average values, accounting for 92 or 85 % of daily $p\mathrm{CO_2}$ variations, respectively, as compared with high-frequency $p\mathrm{CO_2}$ measurements at 10-min intervals (Figure 8). Inland waters with low risk of biofouling can withstand an extended monitoring duration between maintenance visits by several months, if temporal resolution is set at hourly scales, considering the trade-off between time resolution and increasing power demand.

## 5 Summary and implications

The brief literature review identified increasing applications of various automated equilibration systems to continuous monitoring of $p\mathrm{CO_2}$. Spray- and marble-type equilibrators have been used successfully to investigate spatial variations in $p\mathrm{CO_2}$ in large rivers and estuaries. Although the measurement accuracy of the two different types of equilibrators have been validated individually by concomitant headspace equilibration measurements or pH/alkalinity-based calculations, more systematic comparisons are required to establish site-specific response times, long-term stability of measurement accuracy, and maintenance requirements. Membrane-enclosed $\mathrm{CO_2}$ sensors have demonstrated to be useful in long-term, high-frequency $p\mathrm{CO_2}$ monitoring in headwater streams and lakes, yet few studies have used them in large rivers and estuaries or in aquatic systems severely impacted by human influence. Recent advances in coupling equilibrators or membrane-based equilibration devices with the analysis of GHGs other than $\mathrm{CO_2}$ and C isotopes are promising for broader applications of the



tested equilibration systems. However, these will demand further field tests for long-term deployments in various inland

water systems.

In the laboratory tests and field tests at 12 sites, the $pCO_2$ measurements of the three tested equilibration systems agreed well with each other and the headspace equilibration measurements. Both the discrete measurements at 12 sites and underway measurements along the river reach were highly variable in $pCO_2$ but demonstrated rapid response rates and accuracy for the two tested equilibrator types. These results, combined with clogging and power supply problems arising

from long-term deployment of equilibrators, suggest that equilibrators are better suited for relatively short underway measurements than long-term deployment. The membrane-enclosed sensor exhibited longer response times than the equilibrators, especially at low water flow, representing a disadvantage in observing rapid, large $pCO_2$ variations. Nevertheless, this sensor captured large spatial variations of $pCO_2$ within its upper detection limit during the transect study along the highly urbanized river reach. Copper-mesh screening was efficient in reducing inaccuracy of $pCO_2$ measurements

attributed to biofouling on the membrane surface as a result from a long deployment in eutrophic water. We suggest that the membrane-enclosed sensor can be used for both underway and long-term continuous measurements if the sensor has a proper detection range and can be protected by a biofouling-resistant covering.

To provide a framework for different approaches employed in addressing high variability of $pCO_2$ in inland waters, the dimensions of spatiotemporal variability of $pCO_2$ observation were characterized based on the concept of scale and

categorized according to extent and resolution in Figure 9 (Wiens, 1989; Schneider et al., 2001). The first and second dimensions represent the spatial and temporal extent, respectively, as adopted in the traditional space-time scale analysis (Schneider et al., 2001). The third dimension connotes the resolution of the given temporal and spatial extent, which is generally represented by measurement frequency ranging from discrete sampling with low frequency to continuous measurement with high frequency. Continuous underway measurements provide the data that are spatially fine in the

resolution and broad in the extent, but limited in temporal extent. Long-term continuous measurements produce temporally fine and broad data, but are limited in spatial extent. Repeated discrete samplings at multiple sites could address the spatiotemporal variability of $pCO_2$, probably covering a broader spatial extent, but at a coarser resolution. Although inland water $pCO_2$ studies are advancing toward fine–resolution, broad-extent observation, no single approach can fully unveil the high spatiotemporal variability. As the multidisciplinary approach of macrosystems ecology calls upon coordinated multiple

approaches for appreciating the spatiotemporal variability in complex systems (Levy et al., 2014), the results shown here to indicate limitations of individual monitoring methods suggest that $pCO_2$ studies in river systems under strong human influence establish a three-pronged approach: coordinated monitoring involving repeated discrete samplings at multiple sites, a long-term monitoring in a few selected sites, and continuous underway measurements along river reaches that are highly variable in $pCO_2$. To better constrain both natural and anthropogenic factors that determine spatiotemporal dynamics of $CO_2$

in diverse inland water systems, equilibration systems need to resolve the high variability of $pCO_2$ across space and time. Although the accuracy of the tested equilibration systems have been validated by our tests and other studies, they are still limited in their applicability to long-term deployments under difficult field conditions such as limited power supply and





biofouling. Our technical recommendations and caveats can form a solid empirical basis for further studies required to improve the performance and maintenance of equilibration systems during continuous $pCO_2$ monitoring in inland waters.

**Acknowledgements**

This work was supported by the National Research Foundation of Korea funded by the Korean Government (2014R1A2A2A01006577). We thank Borami Park and Most Shirina Begum for their assistance with fieldwork and sample analysis. Special thanks go to Gwenaël Abril, David Butman, Will Gagne-Maynard, Mark Johnson, Peter A. Raymond, Jeffrey Richey, and Enrique Sawakuchi for providing us with information on their equilibration systems. We gratefully
acknowledge the logistical support provided by the Hangang Project Headquarters, Seoul Metropolitan Government.

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



**Table 1. Summary of three investigated equilibration methods for inland water $pCO_2$ measurement.**

| System | Principle | Equilibration method | References |
|---|---|---|---|
| Headspace equilibration | Manual; active | Manual shaking for a given time equilibrates gas concentrations in headspace air and the water collected in a bottle or syringe. | Kling et al., 1992; Hope et al., 1995 |
| Marble-type equilibrator Spray-type equilibrator | Automatic; active | Gas equilibration between air and water is facilitated by enhanced gas exchange over the surface of marbles or by spraying gas-containing water droplets. | Freely et al., 1998; Frankignoulle et al., 2001; Abril et al., 2006 |
| Membrane-enclosed sensor | Automatic; passive* | A water-impermeable, gas-permeable membrane (e.g., PTFE) allows gas equilibration between the inside and outside of the membrane. | Johnson et al., 2010 |

* Passive equilibration based on gas diffusion





**Figure captions**

**Figure 1.** Manual headspace equilibration and three automated equilibration systems for water $pCO_2$ measurement. Blue and grey arrows indicate the flow of water and gas, respectively. Dashed grey arrows denote the equilibration of $pCO_2$ over the air-water interface.

**Figure 2.** Laboratory cross-validation tests of $pCO_2$ equilibration systems. The headspace and membrane-enclosed sensor
were tested using deionized water exposed to a range of $CO_2$ concentrations (a). Response time was compared between the membrane-enclosed sensor, and marble- and spray-type equilibrators (b). The dashed, grey lines indicate the range of $pCO_2$ in tap water used during the test.

**Figure 3.** Comparison of $pCO_2$ measurements by the headspace equilibration, membrane-enclosed sensor, and spray-type equilibrator at 12 sites spanning from a forested headwater stream (site 1) to the estuary of the Han River (site 12). The site
numbers follow the order from the most upstream to the most downstream site. Grey thick and thin barcodes represent a dam and a weir, respectively, the distance from which, together with the specific flow condition during field measurements, were used to distinguish flowing and standing water types. Asterisks indicate the significance of the coefficient of variation calculated from the compared measurements (*: < 10 %, **: < 5 %). Note that the Y-axis is log-transformed.

**Figure 4.** Relationships between the response time ($t_{100}$) and $\Delta pCO_2$ determined as the difference between the initial and
stabilized final $pCO_2$ measurement for a spray-type equilibrator (a) and a membrane-enclosed sensor (b). Note that the X-axis has a log scale.

**Figure 5.** Continuous underway measurements of $pCO_2$ using a spray-type equilibrator, marble-type equilibrator, and a membrane enclosed-sensor along an upper estuarine reach of the Han River receiving loads of organic matter, inorganic nutrients, and $CO_2$ from urban streams indicated by brown arrows. The cruise speed was maintained around 10 km h$^{-1}$ over
the distance of 10 km. Measurements of $pCO_2$ using headspace equilibration (yellow circle) were performed on board. Note that the membrane-enclosed sensors did not capture drastic increases in $pCO_2$ after noon due to the upper detection limit of the sensors (approximately 7,000 and 10,000 ppm for each, grey arrow).

**Figure 6.** Continuous $pCO_2$ measurements at an upper tidal reach of the Han River using a membrane-enclosed sensor without ("Membrane sensor") and with copper mesh screening ("Membrane+Cu sensor") (a). Gray arrows indicate the times
biweekly maintenance was conducted. Relative difference of $pCO_2$ was calculated from the natural log-transformed ratio between values of the bulk and the copper mesh-wrapped sensors (b). Relationship between day-averaged $pCO_2$ values using the "membrane sensor" and the "membrane+Cu" sensor (c).

**Figure 7.** The relationship between pH and $pCO_2$ during successive 4-day monitoring periods following maintenance (a) and the relationship between coefficient of variations (CVs) of daily means of pH and $pCO_2$ (b). Curves are from LOESS (locally
weighted scatterplot smoothing) fitting.

**Figure 8.** Temporal resolution effects on $pCO_2$ measurements using a membrane-enclosed sensor with a copper mesh covering. The data obtained from the copper mesh-wrapped sensor in Figure 7 are presented by modifying temporal





resolutions from 10 min to 4 h. The bottom panel shows daily mean, minimum, and maximum $p\mathrm{CO_2}$ values normalized to the mean of the four calculations.

**Figure 9.** Conceptual view for spatiotemporal scales of $p\mathrm{CO_2}$ monitoring approaches.




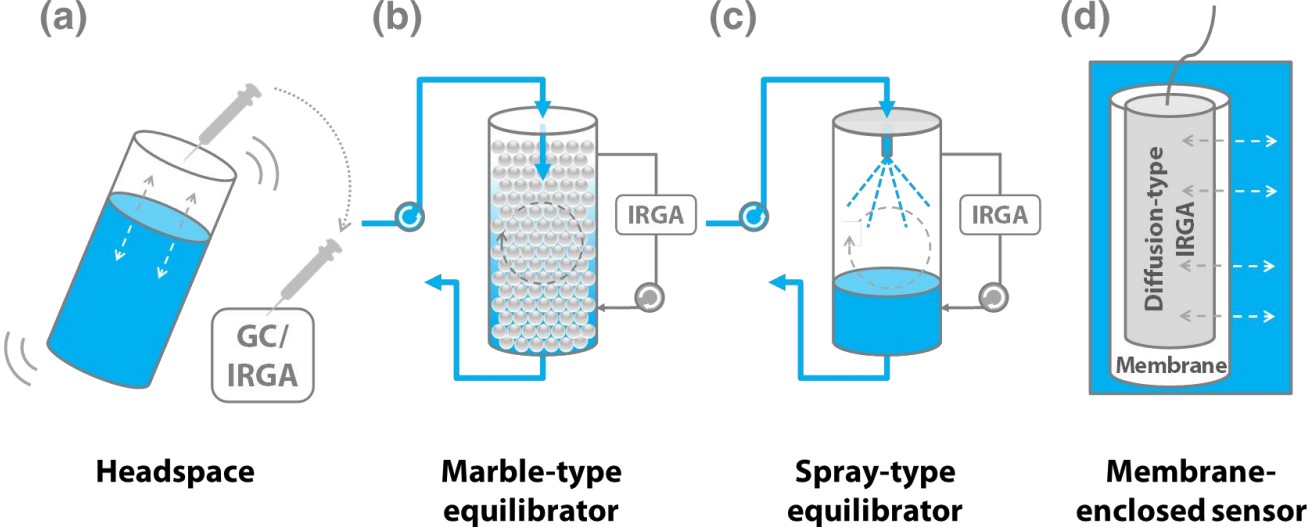

**Figure 1.**





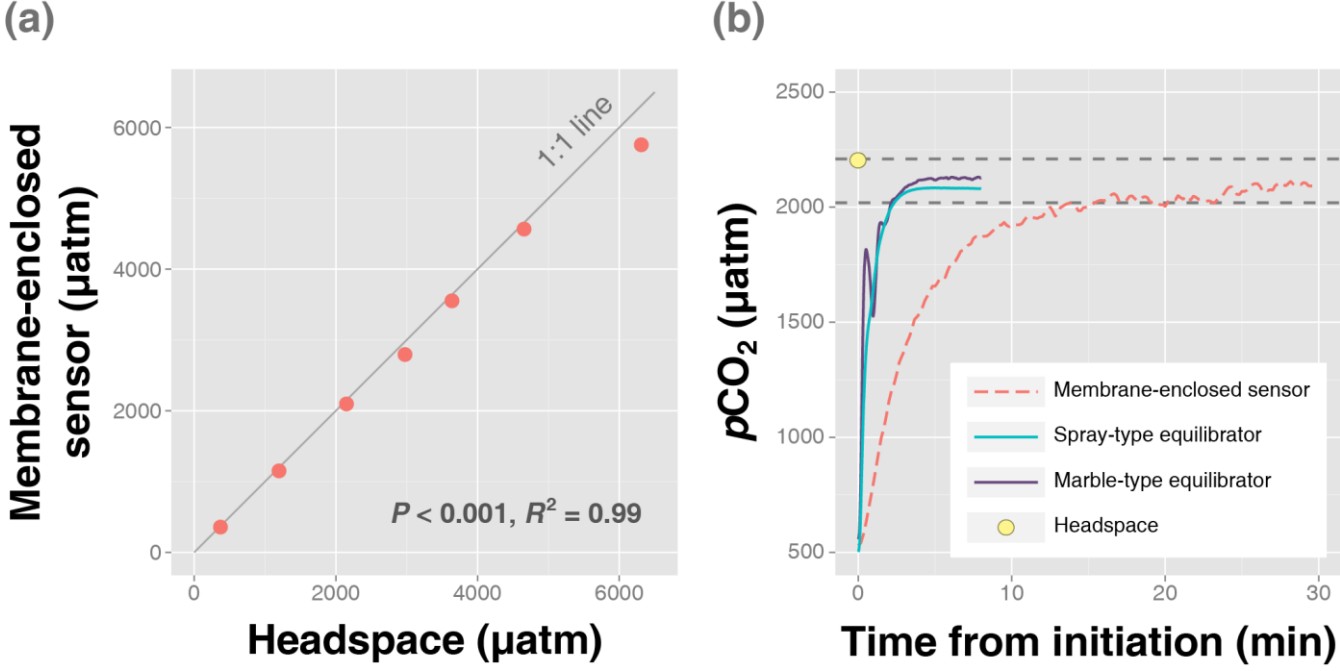


**Figure 2.**





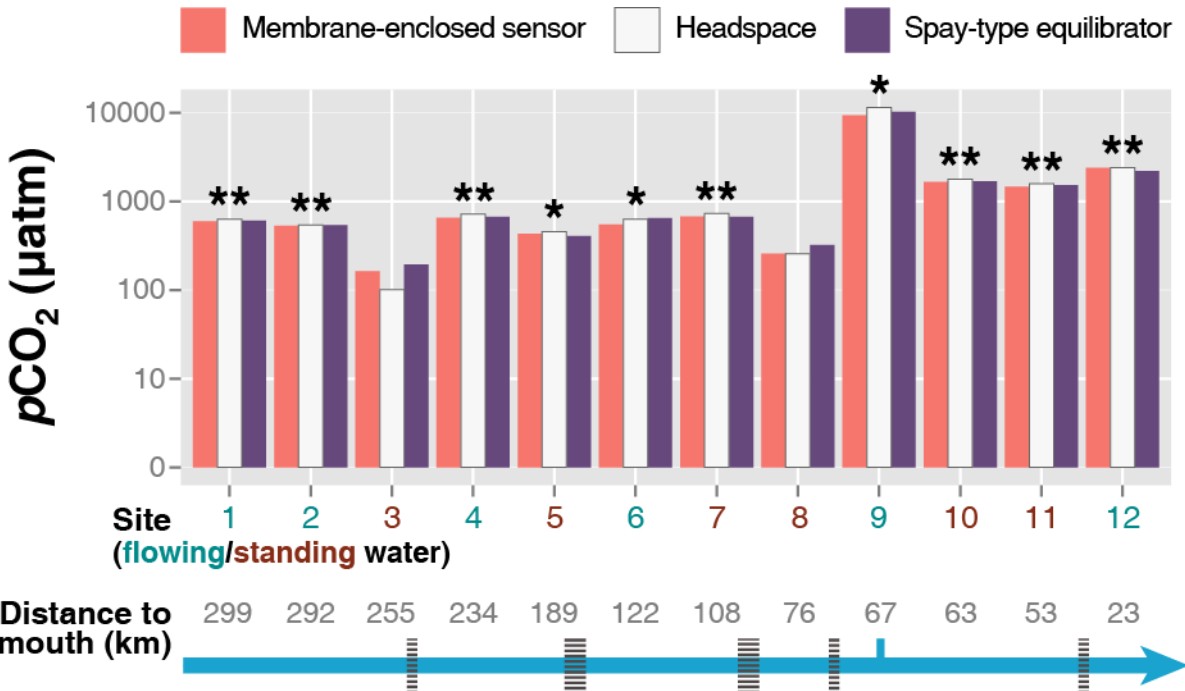

**Figure 3.**





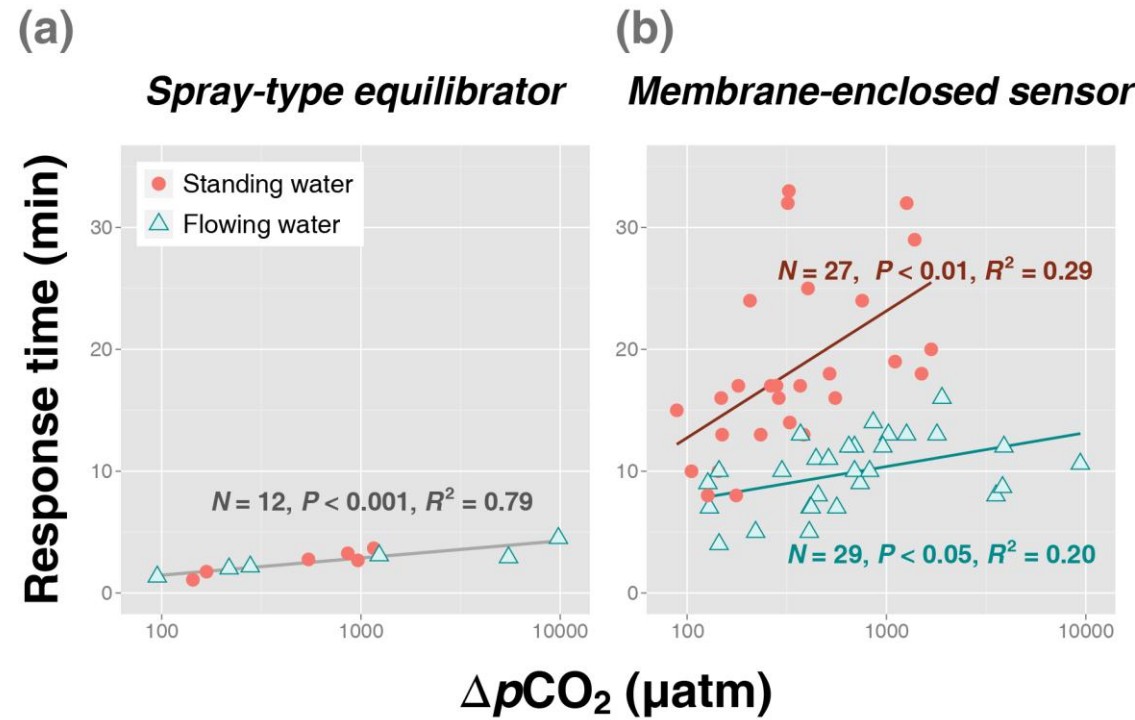

**Figure 4.**





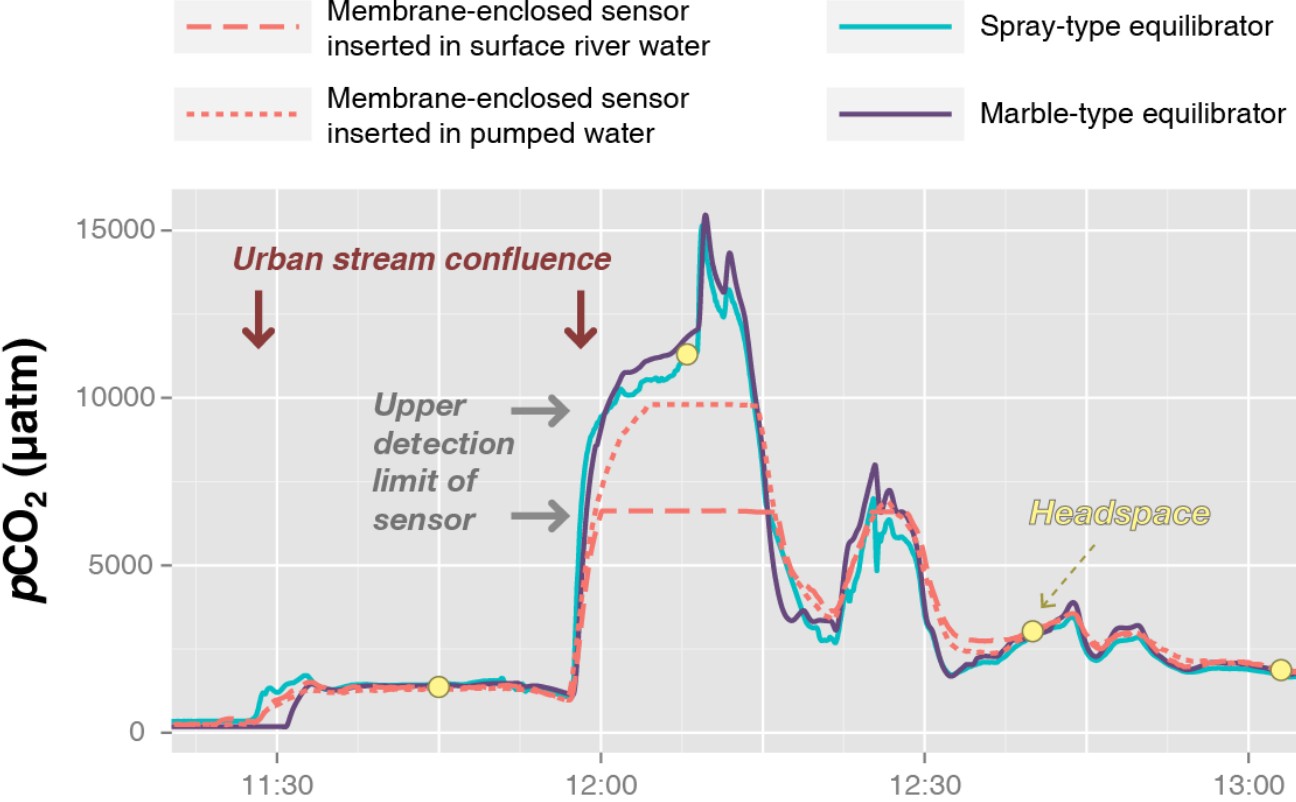


**Figure 5.**





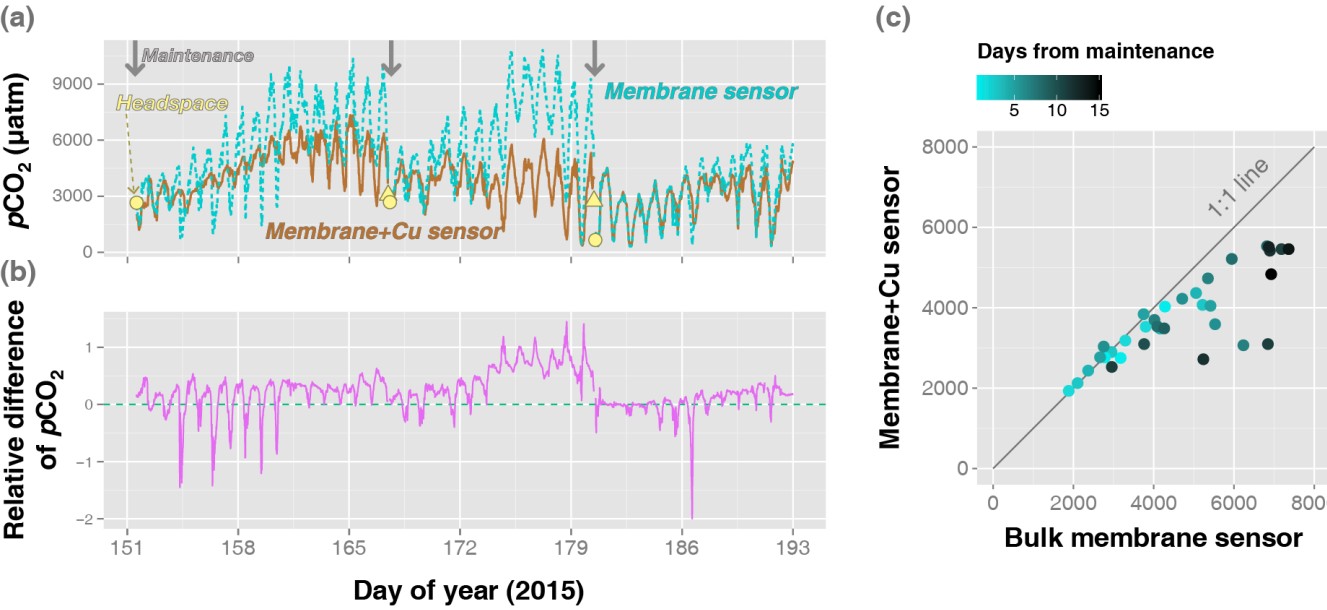

**Figure 6.**





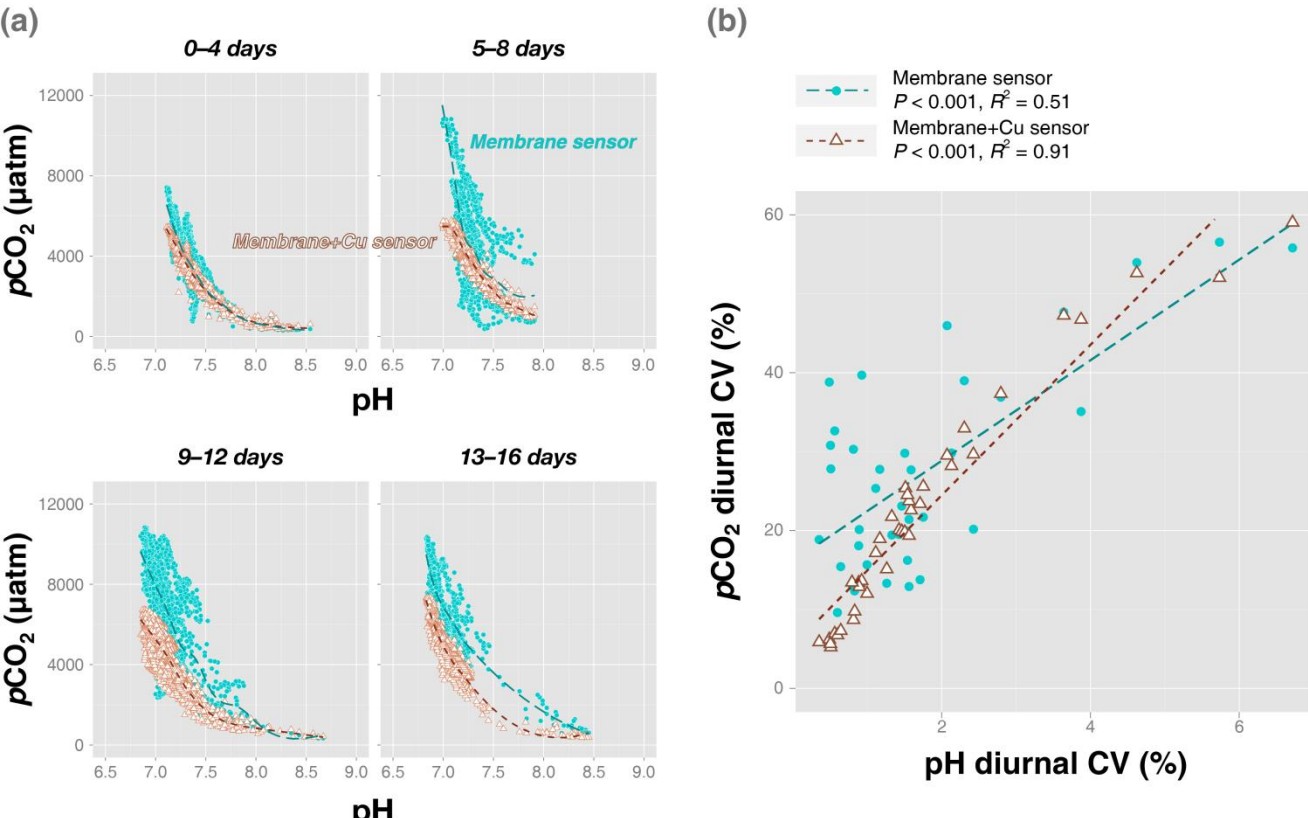

**Figure 7.**



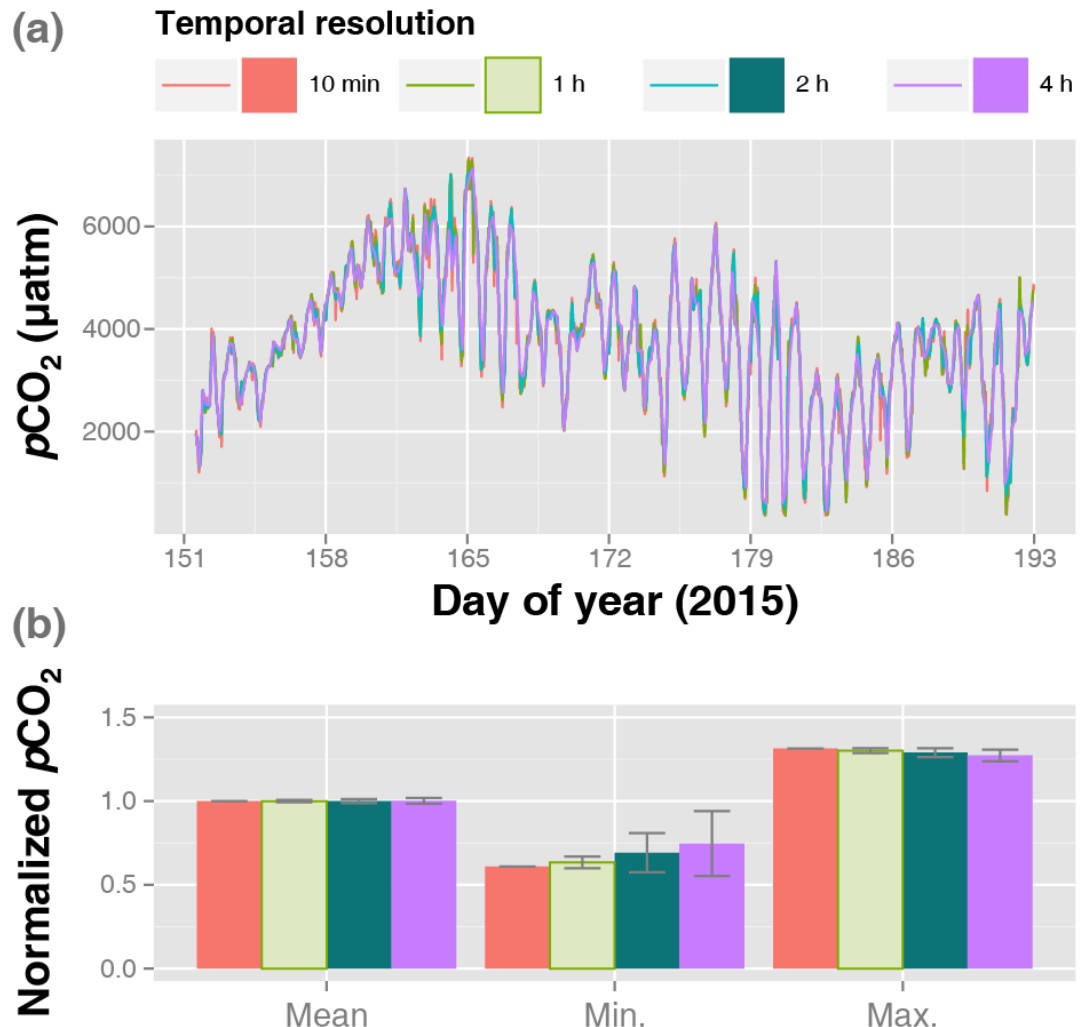

**Figure 8.**





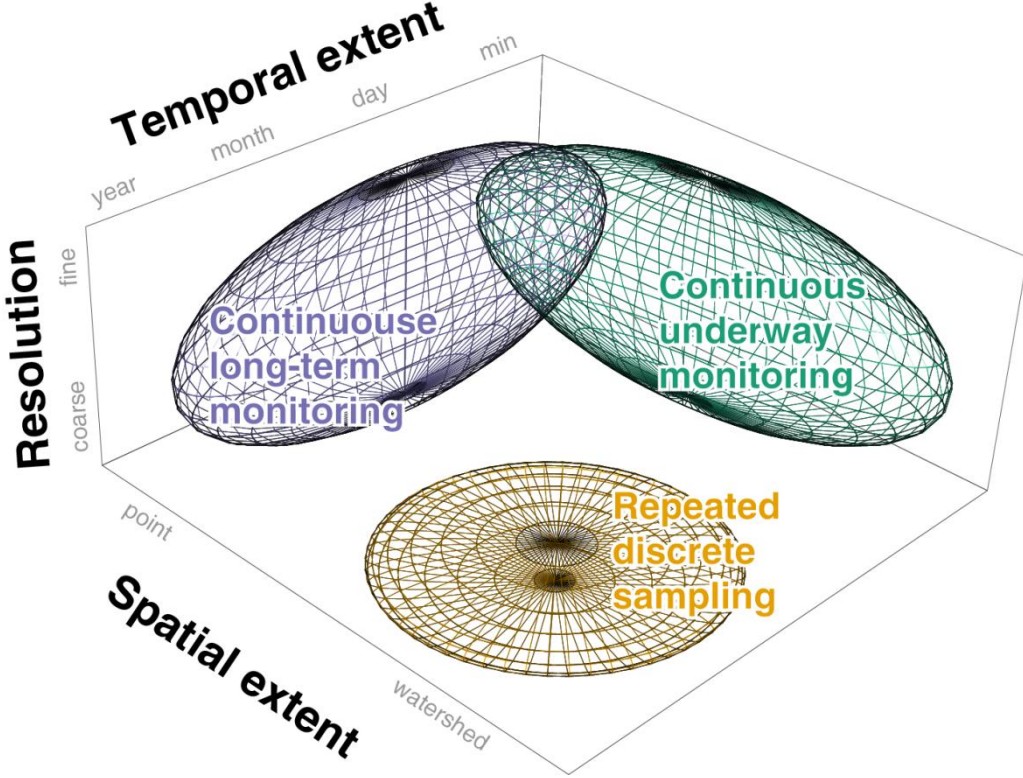

**Figure 9.**