# Peer review of "Technical note: Assessing gas equilibration systems for continuous $pCO_2$ measurements in inland waters"

_Biogeosciences, 2016_

## Referee Comment (RC1) · Anonymous Referee #1 · 21 Mar 2016

Review of Technical note: Applying equilibration systems to continuous measurements of pCO2 in inland waters by Yoon et al.

This paper provides an analysis of 3 commonly used equilibration systems for measurement of water column pCO2 – the spray type "Weiss" equilibrator, the marble equilibrator and a membrane enclosed system. The authors present data from a series of laboratory and field experiments to assess the pros and cons of each system.

The paper claims to be a combine "literature review" and experimental paper, yet the review of the literature is somewhat limited. The title states it relates to "inland waters" but perhaps this is better changed to "freshwater systems" as there are little data from estuarine systems, which have been investigated thoroughly using the techniques

described here. Equilibration systems have been reviewed rather extensively in the past, and the performance of the individual systems assessed here have already been detailed. The paper does present some new information on biofouling with the membranes systems which would be of interest to those using similar techniques.

The paper while generally well-written does require some editing to improve the readability/English (e.g. line15, line 56 what is high leverage of organic acids?, line 136 . . . which is mostly an IRGA. . . etc.).

Specific comments

Introduction CH4 is mentioned at line 30, but nowhere else, I suggest removing this reference as it gives the reader the expectation there will be some discussion about this.

Methods Some more details in the methods would also be helpful. For example was temperature and pressure measured within the marble and spray-type equilibrators, if not were the equilibrators vented to the atmosphere, and how ere temperature differences between the water column and the equilibrator dealt with.

Line 256 0, 500, 5000, 10000 ppm?

Line 269 I do not think one test on response time is adequate to draw too many conclusions – some replication would add some strength to this analysis. Also what about a high to low concentration step – this could take a considerable time in the membrane system. Do the authors have any explanation for the noisy response time data from the marble equilibrator? Also while t 95 and t100 has been used in the past, the best way to assess equilibration time are the models presented by Johnson 1999 [Johnson, J. E. Evaluation of a seawater equilibrator for shipboard analysis of dissolved oceanic trace gases. Anal. Chim. Acta 395, 119-132 (1999)].

Line 303 – Can the authors give some details about how this 10 km/h speed was determined? It seems too fast to assess changes over a 10 km stretch of river (i.e. 1

hour transit time)

Line 322 The reader is initially given the impression that the 3 systems will be compared for the studies – yet the 3 systems are only compared for the survey data. Perhaps this can be clarified earlier, or in the title

Line 344 Was the data corrected for equilibration time in the regression analysis?

Line 350 Can the authors give a bit more detail about what the aim of this analysis is.

Line 380-382 This is also due to the difference in diffusivity between the water-air interface (spray and marble equilibrators) and the water PTFE interface

Line 384 – 391 What about the effect of temperature on diffusivity?

Line 395 I would suspect that allowing only 1 x response time for point measurements would not allow for any changes in the ambient changes in pCO2 during the measurement interval.

Line 401 – To me it looks like the marble equilibrator gives consistently hiogher pCO2 values for the elevated pCO2 areas of the river. Do the authors have an explaination for this? Was pressure measured in the equilibrators? Was temperature measured in the equilibrators? These are very important measurements to make!

Line 408 - Do the authors mean "stationary" rather than discrete measurements (discrete implies headspace measurements)

Line 419 – This has been done in estuaries in recent times, again perhaps expand to include estuaries in the analysis or use more specific terminology rather than inland waters

Line 438-439 Biofouling could cause a shift either way (CO2 increase or decrease) depending upon the community composition.

Figure 3 – I would recommend not using a log scale as this hides some of the differences between equilibrators. Alternatively if the authors add the measured values to the figure (perhaps at 90 degree angle within each bar) that would allow the reader to easily see how the systems compare

―――――――――――――――――――――

---

## Referee Comment (RC2) · Anonymous Referee #2 · 29 Mar 2016

General Comments The paper under review for biogeosciences presents a literature review combined with laboratory and field tests to evaluate the application potential of three widely used automated equilibration systems to continuous long-term or under-way pCO2 measurements. The paper is generally well-written and easy to follow, but I found some grammar and sentence structure issues and I am not a native speaker. I also found some inconsistences between the figure and the results descriptions (see specific comments). In addition, some objectives of the study were not achieved.

As you wrote "This study aims to review advantages and disadvantages of widely used pCO2 equilibration methods and automated equilibration systems that can be used for continuous monitoring of highly variable pCO2 across". The "short review" of this

paper with the advantages and disadvantages of widely used pCO2 equilibrators is not a novelty for continuous aquatic pCO2 measurements. The studies of Santos et al. 2012 and Webb et al. 2016 (including others) presents laboratory step experiments on six different equilibrators to constrain CO2 equilibration time constants and short reviews of the equilibration technique, including shower-head, marble and membrane type equilibrators.

I think you must focus on the new information that this paper provide about improvements in the aquatic pCO2 measurements, which are the long-term deployment of the equilibrators under various field conditions and biofouling with the membrane systems. You must to describe the equilibrator systems with details (the systems were poorly described). The figure 1 and the text did not present details of the measurement-systems, and I think this is very important. In addition, some tests were not performed for the marble equilibrator. I think that is important provide one or two tables with the field and laboratory test results.

You compare the drifts of the pCO2 results for the membrane equilibrators comparing the relationship between pH and pCO2 during successive 4-day monitoring periods following maintenance. However, pH measurement method is missing in the Method section. Since it is used to evaluate the reliability of measured pCO2, it must be evaluated more rigorous. Apparently, the problematic of long-term monitoring of pCO2 is still unsolved (the drifts of the results are very high if is not applied continuous maintenance of the measuring system).

Specific Comments

Line 11: Replace for emissions.

Line 18: ''. . .upper detection limit of the sensor''. What is this limit?

Line 17: The overall results suggest that the equilibrators are better suited for relatively short underway measurements than long-term deployment. Why? Do you have suggestions to improve the equilibration systems in order to long-term pCO2 monitoring? I think you must discuss better this point.

Line 26: First sentence confuse.

Line 26: I think "emission" or "degassing" is better than evasion.

Line 27: I think the "respiration" is more adequate.

Please review all the references. I found some mistakes.

Line 30: You wrote CO2 and CH4. I think is better write dioxide carbon (CO2) and methane (CH4).

Line 35 and 36: Confused. You must explain better the principles of direct and indirect measurements. This sentence is not clear and not sufficient.

Line 36: "... between water and air and gas transfer..."? This is not clear, please rewrite.

Line 39-44: Please read and include information of Lorke et al. (2015) paper. There are important considerations about the floating chamber measurements and improvements on this technique to application for running waters.

Line 50: from pCO2 measurements.

Line 51 and 52: You can also calculated pCO2 from dissolved inorganic carbon (DIC) and total alkalinity (TA) and ancillary parameters. Please include this information. You can read Dickson (2010) to include more accurate statements about the indirect calculations of pCO2.

Lines 61 and 62: SOCAT?

Line 70: delete "from polluted waterways" ?

You did not present the results of the tests (you must insert one or two tables with the results of the field and laboratory tests).

Figure 1: This information is not sufficient. Please, provide more details about the measurement systems of pCO2. For example, see the Figure 1 in Frankignoulle et al. (2001).

As you stated that "Our review and cross-validation tests focus on three automated equilibration systems: spray- and marble-type equilibrators and a membrane-enclosed sensor (Table 1)" you must provide more details about the functioning and details of these three systems.

Line 121 – 124: Bakker et al. (1996) measuring pCO2 in estuarine waters, found "Frequent blockage of the showerhead of the equilibrator with algal material", adding some problems to the measurements. I would like to see some discussion about this problem with the equilibrators.

Lines 129 – 138: A figure with more details of the systems can better elucidate this section of the paper.

Lines 150-152: Do you have some suggestions to turn the equilibrations systems (marble type and showerhead) more automated for long-term monitoring? Please discuss possible improvements that are necessary for long-term monitoring.

Line 154-156: Again, here add one figure can better illustrate how is the passive membrane CO2 equilibration systems, providing details for easy reproducibility.

Line 159 – 161: "There are a small number of commercially available membrane-enclosed sensor 160 systems (e.g., eosGP, Eosense Inc., Canada; Mini-Pro CO2, Pro-Oceanus Systems Inc., Canada)". What are the lower and upper detection limits of these sensors? They can be applied in aquatic systems where the pCO2 values can easily be higher than 10,000 ppmv?

Lines 220-222: A range of 2000 ppmv is "high" in inland waters. Then, this type of system (membrane-enclosed senor) could not be used in some environmental conditions. In addition, I would like to see some discussions about these overestimations.

Lines 233-235: This information is not sufficient. You must provide details of the instruments.

Line 242 – 245. Are you sure that the unit is "mm"? One acrylic tube with this measure is very small, and I think cannot be filled with glass marbles. For example, in Frankignoulle et al. (2001) the vertical Plexiglas measures were: height 80 cm; diameter 10 cm.

Line 250-255: Provide a detailed picture of the complete system.

Line 252: Despite the fact that Johson et al. (2010) provided details of the membrane-enclosed sensor, this is not sufficient for publish in biogeosciences. Your work must yield descriptions of the equilibration systems, both in text and in figures. Your third objective was "to compare the accuracy and maintenance requirements of three selected equilibration systems (a spray- and a marble-type equilibrators and a membrane-enclosed $CO_2$ sensor) for field applications in a series of laboratory and field cross-validation tests". I think that your objective is not just this, rather, I think that is also describe with details these three selected equilibration systems.

Line 256—257: "The $CO_2$ analyzers and sensors were calibrated in the laboratory using $CO_2$ gases of known concentrations (0, 500, 500, and 10,000 ppm) immediately before each laboratory or field test." Why two concentrations of 500? Did you make the calibration after the field test to see the drift of the sensors?

Line 262: About the laboratory test, why you did make the first test just to the membrane-enclosed sensor? This test is not well explained, please rewrite.

Line 275: you did not perform the field test for the marble-type equilibrator. One of your objectives was not realized, since you compared the field tests just for 2 equilibrators. Despite the fact that you assumed that both types would exhibit similar results based on the laboratory results, this cannot be true in field conditions.

Line 295: "The response time was determined as the full time (t100) or 95% of the full

time (t95) it took to a final pCO2 level that represents pCO2 values exhibiting 295 less than 1 % of coefficient of variation (CV) for 2 min." The full time (t100) is unusual for calculations of equilibration time.

Line 303: How were the prior tests of boat speed effect?

Line 314: I think the upper detection limit of the membrane sensor are low and cannot be applied in several inland waters where the natural variations of pCO2 are very higher than these limits.

Line 320-322: This section is confuse. Please rewrite. What preliminary tests did you perform? Why the power supply and air flow dehydration were easiest for the spray-type equilibrator?

Line 324: Why you did not test the marble-type equilibrator for the long-term measurements?

Line 338-342: Move to results section.

Line 345: Did you test the normality of data set? If not follow a normal distribution, you cannot apply the t-test. You must apply the non-parametric tests as Wilcoxon, for example.

Line 349: What are the pH-pCO2 relationships? Not clear in the text. Since it is used to evaluate the reliability of measured pCO2, it must be evaluated more rigorous.

Lines 363 – 365: The results in figure 2 shows that for low pCO2 values the coefficient of variation calculated from the compared measurements were higher than 10%. However, in the text you not explain why this occurs for low pCO2 values.

Line 364: "The CV values were smaller than 5% at all sites except site 3 and 8...". The graph did not show this. Sites 1, 2, 4, 7, 10, 11, 12 CV < 10%. Sites 5, 6, 9 CV < 5%. Sites 3 and 8 CV > 10%. For only 3 sites the CV values were smaller than 5 %, please correct.

Lines 373-374: "The response time of the spray-type equilibrator falls within the usual range of response times reported for the spray- (8 min; Santos et al., 2012)..." Not really. You tests were approximately 4 time more rapid than that reported by Santos et al., 2012 for the spray-type equilibration. You should point some suggestions to explain this difference. Why you did not perform the equilibration time test for the marble-type equilibrator? Where are the results?

Line 384: The figure 4 not showed a logarithm curve, rather, showed a linear tendency.

Line 385: "...with steeper increases observed for the membrane-enclosed sensor, particularly in flowing waters." Is not the contrary? The steeper increase seems to be to the standing waters (Figure 4; red circles).

Line 394: and expected range of pCO2 levels.

Line 405-407: "although it failed to respond to rapid pCO2 increases from the relatively low value at the confluence (11:57) to the concentration peak (12:25) due to the limited detection range." Not just this. If you look at 12:15 and at 12:30, the deviation seems substantial also when the pCO2 values decreased abruptly. You just discussed the deviation when the pCO2 rise, and not when the pCO2 decrease.

Line 430 - 432: What is the explanation to this drift? Not explained in text.

Lines 432-434: "The duration during which relative differences of day-averaged pCO2 between the two sensors..."? I did not understand this section.

Line 436: How you measure pH? What is the accuracy of the method? As you used the relationship pH-pCO2 to examine the increasing biofouling effects with progressing time following the maintenance day, you must provide this information. Also, you need to show that the pH sensor not drift with time.

Line 438: How the biofouling can produced additional CO2 molecules? Explain in the text the process.

Line 442-443: You pointed that "the method validation would require concomitant pCO2 measurements using other equilibration methods". You had all the possibilities to validate this method, but not did, i.e., you had large pCO2 variations and you had three equilibrator types.

Line 446: "Repeated maintenance visits at short intervals of 3 – 5 d may be required for a long term deployment of the sensor without antifouling measures in an inland water site with high levels and large diurnal fluctuations of pCO2." This is difficult depending of the study are. Do you have other suggestions?

Line 460: Interesting result. Can you plot the graph showing these results for oligotrophic waters?

Table 1. Insert one column with the equilibration time for each method.

For figures 2, 3, 4, 5, 6, 7, and 8 I think is better a white fill, without grades and with black contours.

References:

Bakker, D. C. E., De Baar, H. J. W. and De Wilde, H. P. J.: Dissolved carbon dioxide in Dutch coastal waters, Mar. Chem., 55(3-4), 247–263, doi:10.1016/S0304-4203(96)00067-9, 1996.

Dickson, A. G.: The carbon dioxide system in sea water: equilibrium chemistry and measurements, Guide for Best Practices in Ocean Acidification Research and Data Reporting, Office for Official Publications of the European Union, Luxembourg, in press., 2010

Frankignoulle, M., Borges A., and Biondo R.: A new design of equilibrator to monitor carbon dioxide in highly dynamic and turbid environments, Water Res., 35, 344–347, 2001.

Lorke, A., Bodmer, P., Noss, C., Alshboul, Z., Koschorreck, M., Somlai-Haase, C.,

Bastviken, D., Flury, S., McGinnis, D. F., Maeck, a., Müller, D. and Premke, K.: Technical note: drifting versus anchored flux chambers for measuring greenhouse gas emissions from running waters, Biogeosciences, 12(23), 7013–7024, doi:10.5194/bg-12-7013-2015, 2015.

Santos, I. R., Maher, D. T. and Eyre, B. D.: Coupling automated radon and carbon dioxide measurements in coastal waters., Environ. Sci. Technol., 46(14), 7685–91, doi:10.1021/es301961b, 2012.

Webb, J. R., Maher, D. T. and Santos, I. R.: Automated, in situ measurements of dissolved CO2, CH4, and $\delta$13C values using cavity enhanced laser absorption spectrometry: Comparing response times of air-water equilibrators, Limnol. Oceanogr. Methods, (March), doi:10.1002/lom3.10092, 2016.

---

## Author Comment (AC1) · 15 Apr 2016

Please refer to the attached supplement file for our detailed responses to the comments offered by reviewer 1.

Response overview: We appreciate your constructive comments and suggestions. We will revise our manuscript incorporating all your comments and suggestions. The revised manuscript will be checked again by a native English editor to improve accuracy and readability. Some of your and another reviewer's major comments are overlapped, so the same overview of our common responses is provided as below.

(1) Review: There was a common critique on the novelty of our literature review; the

[Figure]

review was evaluated as "somewhat limited" or "not a novelty". We agree to your comment that equilibrator systems have been reviewed and assessed in other studies, but we would like to ask your attention to the fact that our review is the first effort that compares application potentials of the three gas equilibration systems for both underway and temporally continuous pCO2 measurements. To our knowledge, there have been rare efforts to review the three systems from theory to applications focused on freshwater systems. For examples, excellent assessments by Santos et al., (2012) and Webb et al., (2016) focused on the response time of various equilibration systems using laboratory experiments but lacked details on theoretical/technical backgrounds, power requirements, maintenance, and so on. We expect that this introductive review would help researchers initiating pCO2 monitoring study obtain both theoretical and practical information. However, if the editor and reviewers want us to remove or reduce the review section, we will follow the suggestion; we could incorporate the essential contents into the introduction section or keep only focal review components (e.g., applications of gas equilibration systems to continuous measurements) in a separate, but reduced review section.

(2) Additional monitoring data: In response to the comments on the lack of measurements by the marble-type equilibrator in comparing the performance of the three equilibration systems, we will include additional field measurements that would be useful when comparing the performance (e.g., response time) of the three systems.

(3) Methodological details: More detailed descriptions on our gas equilibration systems, together with other in-situ measurements such as pH, and analytical procedures and QC procedures, will be added in the Methods section, Table 1, and Figure 1.

(4) Target water systems: We used inland waters in the title because we also considered estuarine waters in literature review and our field study. For example, our study site includes a tidal reach of the Han River estuary (e.g., sites 10–12 in Figure 3 where underway investigation and long-term monitoring were conducted). We would like to keep this term, but will switch to "freshwater" if the editor and reviewers want us to

focus on freshwater systems.

Please also note the supplement to this comment:
http://www.biogeosciences-discuss.net/bg-2016-54/bg-2016-54-AC1-supplement.pdf

[Figure]

**Supplement:**

**Referee #1**

**Major comments**

This paper provides an analysis of 3 commonly used equilibration systems for measurement of water column $pCO_2$ – the spray type "Weiss" equilibrator, the marble equilibrator and a membrane enclosed system. The authors present data from a series of laboratory and field experiments to assess the pros and cons of each system.

The paper claims to be a combine "literature review" and experimental paper, yet the review of the literature is somewhat limited. The title states it relates to "inland waters" but perhaps this is better changed to "freshwater systems" as there are little data from estuarine systems, which have been investigated thoroughly using the techniques described here. Equilibration systems have been reviewed rather extensively in the past, and the performance of the individual systems assessed here have already been detailed. The paper does present some new information on biofouling with the membranes systems which would be of interest to those using similar techniques.

The paper while generally well-written does require some editing to improve the readability/English (e.g. line15, line 56 what is high leverage of organic acids?, line 136 . . .which is mostly an IRGA. . . etc.).

- **Response overview:** We appreciate your constructive comments and suggestions. We will revise our manuscript incorporating all your comments and suggestions. The revised manuscript will be checked again by a native English editor to improve accuracy and readability. Some of your and another reviewer's major comments are overlapped, so the same overview of our common responses is provided as below.

  **(1) Review**: There was a common critique on the novelty of our literature review; the review was evaluated as "somewhat limited" or "not a novelty". We agree to your comment that equilibrator systems have been reviewed and assessed in other studies, but we would like to ask your attention to the fact that our review is the first effort that compares application potentials of the three gas equilibration systems for both underway and temporally continuous $pCO_2$ measurements. To our knowledge, there have been rare efforts to review the three systems from theory to applications focused on freshwater systems. For examples, excellent assessments by Santos et al., (2012) and Webb et al., (2016) focused on the response time of various equilibration systems using laboratory experiments but lacked details on theoretical/technical backgrounds, power requirements, maintenance, and so on. We expect that this introductive review would help researchers initiating $pCO_2$ monitoring study obtain both theoretical and practical information. However, if the editor and reviewers want us to remove or reduce the review section, we will follow the suggestion; we could incorporate the essential contents into the introduction section or keep only focal review components (e.g., applications of gas equilibration systems to continuous measurements) in a separate, but reduced review section.

  **(2) Additional monitoring data**: In response to the comments on the lack of measurements by the marble-type equilibrator in comparing the performance of the three equilibration systems, we will include additional field measurements that would be useful when comparing the performance (e.g., response time) of the three systems.

  **(3) Methodological details**: More detailed descriptions on our gas equilibration systems, together with other in-situ measurements such as pH, and analytical procedures and QC procedures, will be added in the Methods section, Table 1, and Figure 1.

  **(4) Target water systems**: We used inland waters in the title because we also considered estuarine waters in literature review and our field study. For example, our study site includes a

tidal reach of the Han River estuary (e.g., sites 10–12 in Figure 3 where underway investigation and long-term monitoring were conducted). We would like to keep this term, but will switch to "freshwater" if the editor and reviewers want us to focus on freshwater systems.

**Specific comments**

Introduction - $CH_4$ is mentioned at line 30, but nowhere else, I suggest removing this reference as it gives the reader the expectation there will be some discussion about this.

- **Response:** We will remove this sentence and focus on $CO_2$ in the revision.

Methods - Some more details in the methods would also be helpful. For example was temperature and pressure measured within the marble and spray-type equilibrators, if not were the equilibrators vented to the atmosphere, and how were temperature differences between the water column and the equilibrator dealt with.

- **Response:** First of all, more detailed information about the systems will be added in the revision. Because river water was continuously pumped into the equilibrator at 2.5–3.0 L min$^{-1}$, temperature difference between river water and the circulating water in the equilibrator was negligible, as confirmed by temperature measurements in-stream and in circulating water. Although our systems have vents, we closed the vents during field deployment to enhance the system performance based on our preliminary tests. We will briefly mention vent effects and provide additional test results on the vent effect as supplementary information.

Line 256 0, 500, 5000, 10000 ppm?

- **Response:** Yes. The change will be made in the revision.

Line 269 I do not think one test on response time is adequate to draw too many conclusions – some replication would add some strength to this analysis. Also what about a high to low concentration step – this could take a considerable time in the membrane system. Do the authors have any explanation for the noisy response time data from the marble equilibrator? Also while t 95 and t100 has been used in the past, the best way to assess equilibration time are the models presented by Johnson 1999 [Johnson, J. E. Evaluation of a seawater equilibrator for shipboard analysis of dissolved oceanic trace gases. Anal. Chim. Acta 395, 119-132 (1999)].

- **Response:** First, lab tests were conducted several times and only representative results were presented in the manuscript. We can conduct additional lab tests in response to your comment on any change in response time along high-to-low concentration steps. Second, the noisy response might be explained by the lingering effect of the initial parcel of circulating air in the air loop in which $CO_2$ concentration was as low as in the ambient air. The returned parcel of air in the equilibrator chamber might have diluted the headspace $CO_2$ concentration, causing some noise. We could provide additional data that would clarify the noisy response. Third, the response time is usually calculated based on the exponential decay or e-folding curve fitting, as presented in the cited paper. However, the ideal exponential decay curve did not represent the various response patterns in our field test. Therefore, we used our own criteria to determine the response time as the time required for $pCO_2$ to reach a stabilization point, as presented in the manuscript. We will provide more detailed descriptions and discussion of response time determination.

Line 303 – Can the authors give some details about how this 10 km/h speed was determined? It seems too fast to assess changes over a 10 km stretch of river (i.e. 1 hour transit time)

- **Response:** We determined the boat speed based on our own field tests and Crawford et al. (2014). This will be described in more detail in the revision. The entire underway investigation presented in Figure 5 took ~100 min, including the cruise at 10 km h$^{-1}$ and four stops for sampling (10 min × 4 times). To clarify, an explanatory sentence will be added in the revision.

  *The boat was stopped for ~10 min at each of four discrete sampling points.*

Line 322 The reader is initially given the impression that the 3 systems will be compared for the studies – yet the 3 systems are only compared for the survey data. Perhaps this can be clarified earlier, or in the title

- **Response:** As described in the relevant parts of the manuscript, logistic constraints forced us to select one or two equilibration systems because multi-site tests were conducted as part of another monitoring program. We are willing to add more data on the performance of the marble-type equilibrator system in the revised manuscript.

Line 344 Was the data corrected for equilibration time in the regression analysis?

- **Response:** Yes, we used $p$CO$_2$ values at $t_{100}$. We will specify in the revision which data we used for the analysis.

Line 350 Can the authors give a bit more detail about what the aim of this analysis is?

- **Response:** The following sentences will be added.

  *The analysis was based on the assumption that robust pH-pCO$_2$ relationships would be expected from the carbonate equilibrium model if there were no artifact effects such as sensor biofouling. Temporal changes in pH-pCO$_2$ relationships were examined to assess biofouling-induced deviations from the robust pH-pCO$_2$ relationship.*

Line 380-382 This is also due to the difference in diffusivity between the water-air interface (spray and marble equilibrators) and the water PTFE interface.

- **Response:** We will add more discussion on this issue as follows:

  *In addition, the diffusion-type IRGA of the membrane-enclosed sensor generally exhibited longer response times than the flow-through IRGAs of the equilibrator systems. The passive gas transfer to the sensor unit might contribute to the longer response time of the membrane-enclosed sensor. Gas diffusivity across the water-membrane interface can also differ from the diffusivity between the water-air interface within the equilibrator chambers.*

Line 384 – 391 What about the effect of temperature on diffusivity?

- **Response:** Temperature could be one of factors that regulate response time. We did additional analysis on temperature effects, but we could not find any effect because temperature did not vary a lot between sampling sites. Following sentence will be added in the revised manuscript.

  *Temperature could also affect response time, but regression analysis did not show any significant relationship between temperature and response time, probably due to the relatively narrow range of temperature variations among sampling sites.*

Line 395 I would suspect that allowing only 1 x response time for point measurements would not allow for any changes in the ambient changes in $p$CO$_2$ during the measurement interval.

- **Response:** We wanted to emphasize adequate deployment times for a spot measurement, not for continuous measurement during a sustained period of time to describe the temporal changes in the ambient $p$CO$_2$. To clarify, the sentences will be revised. If our response does not satisfy the point of your query, please let us know again.

Line 401 – To me it looks like the marble equilibrator gives consistently higher $p$CO$_2$ values for the elevated $p$CO$_2$ areas of the river. Do the authors have an explanation for this? Was pressure measured in the equilibrators? Was temperature measured in the equilibrators? These are very important measurements to make! .

- **Response**: Please refer to our previous response to the same question about temperature and pressure measurements. Differences in the measurements by the two equilibrators around pCO2 peaks, together with some potential explanations, will be provided in the revision.

Line 408 - Do the authors mean "stationary" rather than discrete measurements (discrete implies headspace measurements)

- **Response:** A new term "spot measurement" will be consistently used through the manuscript to refer to spot measurements of $p$CO$_2$ in comparison to continuous underway or long-term measurements.

Line 419 – This has been done in estuaries in recent times, again perhaps expand to include estuaries in the analysis or use more specific terminology rather than inland waters

- **Response:** We will focus on freshwater systems in the revision and the sentence will be revised as follows:

  *CO$_2$ sensors have recently been used for continuous pCO$_2$ monitoring in some freshwater systems (Johnson et al., 2010; Huotari et al., 2013; Peter et al., 2014; Leith et al., 2015). However, these previous studies have rarely examined spatial variations in pCO$_2$ across a wide range of environmental conditions.*

Line 438-439 Biofouling could cause a shift either way (CO$_2$ increase or decrease) depending upon the community composition.

- **Response:** In response to your comment, the sentence will be revised as follows:

  *If additional CO$_2$ molecules were produced or consumed by biofilms formed on the "membrane sensor", this could disturb the usual pH-CO$_2$ relationship that can be explained by the carbonate equilibrium model (Nimick et al., 2011).*

Figure 3 – I would recommend not using a log scale as this hides some of the differences between equilibrators. Alternatively if the authors add the measured values to the figure (perhaps at 90 degree angle within each bar) that would allow the reader to easily see how the systems compare

- **Response:** The change will be made in the revision according to your suggestion.

---

## Author Comment (AC2) · 15 Apr 2016

Please refer to the attached supplement file for our detailed responses to the comments offered by reviewer 2.

Response overview: We appreciate your constructive comments and suggestions. We will revise our manuscript incorporating all your comments and suggestions. The revised manuscript will be checked again by a native English editor to improve accuracy and readability. Some of your and another reviewer's major comments are overlapped, so the same overview of our common responses is provided as below.

(1) Review: There was a common critique on the novelty of our literature review; the

[Figure]

review was evaluated as "somewhat limited" or "not a novelty". We agree to your comment that equilibrator systems have been reviewed and assessed in other studies, but we would like to ask your attention to the fact that our review is the first effort that compares application potentials of the three gas equilibration systems for both underway and temporally continuous pCO2 measurements. To our knowledge, there have been rare efforts to review the three systems from theory to applications focused on freshwater systems. For examples, excellent assessments by Santos et al., (2012) and Webb et al., (2016) focused on the response time of various equilibration systems using laboratory experiments but lacked details on theoretical/technical backgrounds, power requirements, maintenance, and so on. We expect that this introductive review would help researchers initiating pCO2 monitoring study obtain both theoretical and practical information. However, if the editor and reviewers want us to remove or reduce the review section, we will follow the suggestion; we could incorporate the essential contents into the introduction section or keep only focal review components (e.g., applications of gas equilibration systems to continuous measurements) in a separate, but reduced review section.

(2) Additional monitoring data: In response to the comments on the lack of measurements by the marble-type equilibrator in comparing the performance of the three equilibration systems, we will include additional field measurements that would be useful when comparing the performance (e.g., response time) of the three systems.

(3) Methodological details: More detailed descriptions on our gas equilibration systems, together with other in-situ measurements such as pH, and analytical procedures and QC procedures, will be added in the Methods section, Table 1, and Figure 1.

(4) Target water systems: We used inland waters in the title because we also considered estuarine waters in literature review and our field study. For example, our study site includes a tidal reach of the Han River estuary (e.g., sites 10–12 in Figure 3 where underway investigation and long-term monitoring were conducted). We would like to keep this term, but will switch to "freshwater" if the editor and reviewers want us to

focus on freshwater systems.

Please also note the supplement to this comment:
http://www.biogeosciences-discuss.net/bg-2016-54/bg-2016-54-AC2-supplement.pdf

**Supplement:**

**Reviewer #2**

**General Comments**

The paper under review for biogeosciences presents a literature review combined with laboratory and field tests to evaluate the application potential of three widely used automated equilibration systems to continuous long-term or underway  $pCO_2$  measurements. The paper is generally well-written and easy to follow, but I found some grammar and sentence structure issues and I am not a native speaker. I also found some inconsistences between the figure and the results descriptions (see specific comments). In addition, some objectives of the study were not achieved.

As you wrote "This study aims to review advantages and disadvantages of widely used  $pCO_2$  equilibration methods and automated equilibration systems that can be used for continuous monitoring of highly variable  $pCO_2$  across". The "short review" of this paper with the advantages and disadvantages of widely used  $pCO_2$  equilibrators is not a novelty for continuous aquatic  $pCO_2$  measurements. The studies of Santos et al. 2012 and Webb et al. 2016 (including others) presents laboratory step experiments on six different equilibrators to constrain  $CO_2$  equilibration time constants and short reviews of the equilibration technique, including shower-head, marble and membrane type equilibrators.

I think you must focus on the new information that this paper provide about improvements in the aquatic  $pCO_2$  measurements, which are the long-term deployment of the equilibrators under various field conditions and biofouling with the membrane systems. You must to describe the equilibrator systems with details (the systems were poorly described). The figure 1 and the text did not present details of the measurement-systems, and I think this is very important. In addition, some tests were not performed for the marble equilibrator. I think that is important provide one or two tables with the field and laboratory test results.

You compare the drifts of the  $pCO_2$  results for the membrane equilibrators comparing the relationship between pH and  $pCO_2$  during successive 4-day monitoring periods following maintenance. However, pH measurement method is missing in the Method section. Since it is used to evaluate the reliability of measured  $pCO_2$ , it must be evaluated more rigorous. Apparently, the problematic of long-term monitoring of  $pCO_2$  is still unsolved (the drifts of the results are very high if is not applied continuous maintenance of the measuring system).

• **Response overview:** We appreciate your constructive comments and suggestions. We will revise our manuscript incorporating all your comments and suggestions. The revised manuscript will be checked again by a native English editor to improve accuracy and readability. Some of your and another reviewer's major comments are overlapped, so the same overview of our common responses is provided as below.

(1) **Review**: There was a common critique on the novelty of our literature review; the review was evaluated as "somewhat limited" or "not a novelty". We agree to your comment that equilibrator systems have been reviewed and assessed in other studies, but we would like to ask your attention to the fact that our review is the first effort that compares application potentials of the three gas equilibration systems for both underway and temporally continuous  $pCO_2$  measurements. To our knowledge, there have been rare efforts to review the three systems from theory to applications focused on freshwater systems. For examples, excellent assessments by Santos et al., (2012) and Webb et al., (2016) focused on the response time of various equilibration systems using laboratory experiments but lacked details on theoretical/technical backgrounds, power requirements, maintenance, and so on. We expect that this introductive review would help researchers initiating  $pCO_2$  monitoring study obtain both theoretical and practical information. However, if the editor and reviewers want us to remove or reduce the review section, we will follow the suggestion; we could incorporate the essential contents into

the introduction section or keep only focal review components (e.g., applications of gas equilibration systems to continuous measurements) in a separate, but reduced review section.

(2) Additional monitoring data: In response to the comments on the lack of measurements by the marble-type equilibrator in comparing the performance of the three equilibration systems, we will include additional field measurements that would be useful when comparing the performance (e.g., response time) of the three systems.

(3) Methodological details: More detailed descriptions on our gas equilibration systems, together with other in-situ measurements such as pH, and analytical procedures and QC procedures, will be added in the Methods section, Table 1, and Figure 1.

(4) **Target water systems**: We used inland waters in the title because we also considered estuarine waters in literature review and our field study. For example, our study site includes a tidal reach of the Han River estuary (e.g., sites 10–12 in Figure 3 where underway investigation and long-term monitoring were conducted). We would like to keep this term, but will switch to "freshwater" if the editor and reviewers want us to focus on freshwater systems.

**Specific Comments**

Line 11: Replace for emissions.

• **Response:** The "evasion" will be replaced by "emission".

Line 18: "... upper detection limit of the sensor". What is this limit?

• **Response:** We also think that this sentence would be confusing without more detailed descriptions, so it will be rewritten like "...for the river section where  $pCO_2$  varies within the sensor detection range"

Line 17: The overall results suggest that the equilibrators are better suited for relatively short underway measurements than long-term deployment. Why? Do you have suggestions to improve the equilibration systems in order to long-term  $pCO_2$  monitoring? I think you must discuss better this point.

• **Response:** Our suggestions based on additional discussion will be added in the revision. The sentence in the abstract will rewritten as follows:

The overall results suggest that the fast response allows the equilibrator systems to capture large spatial variations in  $pCO_2$  during relatively short underway measurements, while technical challenges such as clogging and desiccant maintenance should be carefully addressed for their long-term deployment over several days to weeks. The membrane-enclosed sensor can be an alternative tool for long-term continuous measurements, if membrane biofouling can be overcome using anti-fouling measures such as copper-mesh covering.

Line 26: First sentence confuse.

• **Response:** The sentence will be rewritten as follows:

The importance of carbon dioxide  $(CO_2)$  emission from inland waters has increasingly been recognized in the global carbon cycle research.

Line 27: I think the "respiration" is more adequate.

• **Response:** The change will be made in the revision (The emission of carbon (C) resulting from the respiration in inland waters...).

Line 26: I think "emission" or "degassing" is better than evasion.

• **Response:** The change will be made in the revision (The emission of carbon (C) resulting from the respiration in inland waters...).

Please review all the references. I found some mistakes.

• **Response:** We will thoroughly check any mistakes in the cited references.

Line 30: You wrote CO2 and CH4. I think is better write dioxide carbon (CO2) and methane (CH4).

• **Response:** The change will be made in the revision.

Line 35 and 36: Confused. You must explain better the principles of direct and indirect measurements. This sentence is not clear and not sufficient.

Line 36: "... between water and air and gas transfer. ... "? This is not clear, please rewrite.

• **Response:** The principles of direct and indirect measurements are presented in separate paragraphs (see Lines 35–49 and Lines 50–61, respectively). To clarify, the sentence will be changed as follows:

The emission of  $CO_2$  can be determined either by directly measuring the transfer of  $CO_2$  across the water-air interface or by estimating the flux based on (1) differences in the partial pressure of  $CO_2$  ( $pCO_2$ ) between water and air and (2) gas transfer velocity.

Line 39-44: Please read and include information of Lorke et al. (2015) paper. There are important considerations about the floating chamber measurements and improvements on this technique to application for running waters.

• **Response:** Thank you for the useful reference. The paper will be cited as follows:

Disrupted natural turbulence inside the floating chamber could cause the risk of overestimating  $CO_2$  flux (Vachon et al., 2010), especially when the chamber is anchored at a fixed spot (Lorke et al., 2015). Lorke et al. (2015) recently proposed improved floating chamber designs that minimize the bias of gas transfer velocity, including a freely drifting chamber on running water and an anchored chamber with a close contact over water surface. However, validation and further technical improvements are still required for practical applications of these proposed chamber systems.

Line 50: from pCO2 measurements.

• **Response:** The change will be made in the revision.

Line 51 and 52: You can also calculated  $pCO_2$  from dissolved inorganic carbon (DIC) and total alkalinity (TA) and ancillary parameters. Please include this information. You can read Dickson (2010) to include more accurate statements about the indirect calculations of  $pCO_2$ .

• **Response:** The change will be made in the revision.

The  $pCO_2$  can also be estimated using two of the three variables in the carbonate equilibrium model - pH, alkalinity, and dissolved inorganic C (DIC) (Lewis et al., 1998; Dickson et al., 2007, 2010).

Lines 61 and 62: SOCAT?

• **Response:** The "Surface Ocean CO2" will be added in the revision.

Line 70: delete "from polluted waterways"?

• **Response:** The change will be made in the revision.

the evasion of  $CO_2$  from "polluted waterways" in urbanized watersheds  $\rightarrow$  the evasion of  $CO_2$  from urbanized watersheds

You did not present the results of the tests (you must insert one or two tables with the results of the field and laboratory tests).

• **Response:** The revised manuscript will contain all necessary data incorporating reviewer comments and suggestions.

Figure 1: This information is not sufficient. Please, provide more details about the measurement systems of  $pCO_2$ . For example, see the Figure 1 in Frankignoulle et al. (2001). As you stated that "Our review and cross-validation tests focus on three automated equilibration systems: spray- and marble-type equilibrators and a membrane-enclosed sensor (Table 1)" you must provide more details about the functioning and details of these three systems.

• **Response:** More detailed descriptions will be added in the revised Figure 1 and Table 1, including the photos of three systems that will be included in the figure or as the supplementary information.

Line 121 - 124: Bakker et al. (1996) measuring  $pCO_2$  in estuarine waters, found "Frequent blockage of the showerhead of the equilibrator with algal material", adding some problems to the measurements. I would like to see some discussion about this problem with the equilibrators.

• **Response:** The paper will be cited in discussing potential sources of system failure due to blockage and contamination.

Lines 129 – 138: A figure with more details of the systems can better elucidate this section of the paper.

• **Response:** Specific features of equilibration systems that we used will be presented in Figure 1 and Table 1 with further details provided in the supplementary information.

Lines 150-152: Do you have some suggestions to turn the equilibrations systems (marble type and showerhead) more automated for long-term monitoring? Please discuss possible improvements that are necessary for long-term monitoring.

• **Response:** Some maintenance and technical recommendations will be provided in the discussion section.

Line 154-156: Again, here add one figure can better illustrate how is the passive membrane  $CO_2$  equilibration systems, providing details for easy reproducibility.

• **Response:** The change will be made in the revision.

Line 159 - 161: "There are a small number of commercially available membraneeenclosed sensor systems (e.g., eosGP, Eosense Inc., Canada; Mini-Pro CO2, Pro-Oceanus Systems Inc., Canada)". What are the lower and upper detection limits of these sensors? They can be applied in aquatic systems where the  $pCO_2$  values can easily be higher than 10,000 ppmv?

• **Response:** The detection ranges can be adjusted by manufacturer, from low range (e.g., 0–1000 ppm) to high range (e.g., 0–20%). The sensors are able to cover the range of *p*CO2 in inland waters when the detection ranges are appropriately set and calibrated. This wide detection range will be mentioned in the same sentence.

Lines 220-222: A range of 2000 ppmv is "high" in inland waters. Then, this type of system (membraneenclosed senor) could not be used in some environmental conditions. In addition, I would like to see some discussions about these overestimations.

• **Response:** The cited paper did not elaborate on potential sources of overestimation in the high  $pCO_2$  range, so we cannot provide any explanation here, but we will discuss in more detail when we evaluate our own measurement results from the membrane-enclosed sensors.

Lines 233-235: This information is not sufficient. You must provide details of the instruments.

• **Response:** Aforementioned more detailed information about the systems will be added in the revision.

Line 242 - 245. Are you sure that the unit is "mm"? One acrylic tube with this measure is very small, and I think cannot be filled with glass marbles. For example, in Frankignoulle et al. (2001) the vertical Plexiglas measures were: height 80 cm; diameter 10 cm.

• **Response:** Yes. We devised a smaller system modified from the original marble-type equilibrator of Frankignoulle et al. (2001). The smaller chamber was filled with smaller marbles (diameter: 10 mm) than the larger 20–30 mm marbles used for the bigger chamber. Other researchers have also used small systems for portability; for instance, Abril et al., (2006) used a chamber of 8 cm in diameter and 60 cm in height. We compared small-sized and original equilibrators in laboratory and we found no significant differences in performance between them. We will note this information in the revision.

Line 250-255: Provide a detailed picture of the complete system.

• **Response:** As mentioned before, the change will be made in the revision.

Line 252: Despite the fact that Johson et al. (2010) provided details of the membrane enclosed sensor, this is not sufficient for publish in biogeosciences. Your work must yield descriptions of the equilibration systems, both in text and in figures. Your third objective was "to compare the accuracy and maintenance requirements of three selected equilibration systems (a spray- and a marble-type equilibrators and a membraneenclosed  $CO_2$  sensor) for field applications in a series of laboratory and field crossvalidation tests". I think that your objective is not just this, rather, I think that is also describe with details these three selected equilibration systems.

• **Response:** Again, we will add more information about the system in text and figure in the revision.

Line 256-257: "The CO2 analyzers and sensors were calibrated in the laboratory using CO2 gases of known concentrations (0, 500, 500, and 10,000 ppm) immediately before each laboratory or field test." Why two concentrations of 500? Did you make the calibration after the field test to see the drift of the sensors?

• **Response:** The mistake of the wrong  $CO_2$  gas concentration will be corrected (500  $\rightarrow$  5,000 ppm). Yes, the sensors were also checked following deployment and we found little drift for the deployment periods from which the presented results were obtained.

Line 262: About the laboratory test, why you did make the first test just to the membrane-enclosed sensor? This test is not well explained, please rewrite.

• **Response:** The sentence will be rewritten to describe the objectives and procedures of the tests in more detail.

Line 275: you did not perform the field test for the marble-type equilibrator. One of your objectives was not realized, since you compared the field tests just for 2 equilibrators. Despite the fact that you assumed that both types would exhibit similar results based on the laboratory results, this cannot be true in field conditions.

• **Response:** We will add more field data obtained using the marble-type equilibrator in the revision. We hope it would satisfy your expectation.

Line 295: "The response time was determined as the full time (t100) or 95% of the full time (t95) it took to a final  $pCO_2$  level that represents  $pCO_2$  values exhibiting less than 1 % of coefficient of variation (CV) for 2 min." The full time (t100) is unusual for calculations of equilibration time.

• **Response:** As described in our response to another reviewer's comment, the response time is usually calculated based on the exponential decay or e-folding curve fitting. However, the ideal exponential decay curve did not represent the various response patterns observed in our field tests conducted in various water types and under various initial conditions. Therefore, we used our own criteria to determine the response time as the time required for  $pCO_2$  to reach a stabilization point, as presented in the manuscript. We will provide more detailed descriptions and discussion of response time determination. This approach allowed us to obtain both  $t_{100}$  and  $t_{95}$ , as we presented both  $t_{95}$  and  $t_{100}$  results in the text. For Figure 4,  $t_{100}$  values were selected because they provided slightly higher  $R^2$  and lower *P*-value for regression fittings than  $t_{95}$ . If you want us to replace  $t_{100}$  with  $t_{95}$  data, we will revise the manuscript accordingly.

Line 303: How were the prior tests of boat speed effect?

• **Response**: We determined the boat speed based on our own field tests and Crawford et al. (2010). Three systems exhibited comparable results until boat speed approaches the selected speed. This will be described in more detail in the revision.

Line 314: I think the upper detection limit of the membrane sensor are low and cannot be applied in several inland waters where the natural variations of  $pCO_2$  are very higher than these limits.

• **Response**: The detection ranges can be adjusted by manufacturer, various detection ranges, from low ranges (e.g., 0–1000 ppm) to high ranges (e.g., 0–20%). The sensor we used and other commercially available sensors can cover the wide range of *p*CO2 in inland waters when the sensors with proper detection ranges are well calibrated. This detail will be specified.

Line 320-322: This section is confuse. Please rewrite. What preliminary tests did you perform? Why the power supply and air flow dehydration were easiest for the spraytype equilibrator?

• **Response**: The sentences will be rewritten as follows:

In preliminary tests the spray-type equilibrator was easier to maintain the desiccant that was used to dehydrate air flow compared to the marble-type equilibrator. Logistic considerations also let us opt for the spray-type equilibrator in spot measurements at 12 sites.

Line 324: Why you did not test the marble-type equilibrator for the long-term measurements?

• **Response**: Our tests with both equilibrators were not successful for the same reason as described for the spray-type equilibrator. In our study sites, electric power supply was not available to run equilibrator systems for long-term periods. This will be mentioned more clearly in the revision.

Line 338-342: Move to results section.

• **Response**: The change will be added in the revision.

Line 345: Did you test the normality of data set? If not follow a normal distribution, you cannot apply the t-test. You must apply the non-parametric tests as Wilcoxon, for example.

• **Response**: Yes, we checked the normal distribution using Shapiro-Wilk test.

Line 349: What are the pH- $pCO_2$  relationships? Not clear in the text. Since it is used to evaluate the reliability of measured  $pCO_2$ , it must be evaluated more rigorous.

• **Response:** Following sentences will be added.

The analysis was based on the assumption that robust pH- $pCO_2$  relationships would be expected from the carbonate equilibrium model if there were no artifact effects such as sensor biofouling. Temporal changes in pH- $pCO_2$  relationships were examined to assess biofouling-induced deviations from the robust pH- $pCO_2$  relationship.

Lines 363 - 365: The results in figure 2 shows that for low  $pCO_2$  values the coefficient of variation calculated from the compared measurements were higher than 10%. However, in the text you not explain why this occurs for low  $pCO_2$  values. Line 364: "The CV values were smaller than 5% at all sites except site 3 and 8. . .". The graph did not show this. Sites 1, 2, 4, 7, 10, 11, 12 CV < 10%. Sites 5, 6, 9 CV < 5%. Sites 3 and 8 CV > 10%. For only 3 sites the CV values were smaller than 5%, please correct.

• **Response**: We double checked and the modified sentence and an additional sentence explaining the large CV observed at two sites will be added as follows:

The CV values were smaller than 5% at seven sites and < 10 % at three sites. At two sites where CV values were > 10, even slight differences in measured values between the three methods compared to the results observed at the other sites resulted in relatively large CVs due to the lower pCO2 values than the atmospheric pCO2.

Lines 373-374: "The response time of the spray-type equilibrator falls within the usual range of response times reported for the spray- (8 min; Santos et al., 2012). . ." Not really. You tests were approximately 4 time more rapid than that reported by Santos et al., 2012 for the spray-type equilibration. You should point some suggestions to explain this difference. Why you did not perform the equilibration time test for

the marble-type equilibrator? Where are the results?

• Additional field data using the marble-type equilibrator will be included in the revision. Following change will be made in the revision.

Mean  $t_{95\%}$  and  $t_{100\%}$  for the spray-type equilibrator was 1 min 31 s and 2 min 36 s, respectively, with no difference between standing and flowing waters. The response time of the spray-type equilibrator was shorter than the response time reported for a spray-type equilibrator (8 min; Santos et al., 2012), but similar to the response times of marble-type equilibrators (2 – 3 min; Frankignoulle et al., 2001; Abril et al., 2014). The difference in the response time might result from various factors including the different levels of pCO2 (~100–5000 µatm in this study vs. > 10,000 µatm in the Santos et al.), equilibrator size (251 vs. 1963 cm-3), length of air circuit, and spray nozzle performance.

Line 384: The figure 4 not showed a logarithm curve, rather, showed a linear tendency.

• **Response**: The curve appear linear because of the log-scaled X axis. To clarify, "note a log scale for the x-axis" will be added in the figure caption.

Line 385: "... with steeper increases observed for the membrane-enclosed sensor, particularly in flowing waters." Is not the contrary? The steeper increase seems to be to the standing waters (Figure 4; red circles)."

• **Response**: Yes, the increase was steeper in the standing waters. The sentence will be corrected in the revision.

Line 394: and expected range of  $pCO_2$  levels.

• **Response**: The change will be added in the revision.

Line 405–407: "although it failed to respond to rapid  $pCO_2$  increases from the relatively low value at the confluence (11:57) to the concentration peak (12:25) due to the limited detection range." Not just this. If you look at 12:15 and at 12:30, the deviation seems substantial also when the  $pCO_2$  values decreased abruptly. You just discussed the deviation when the  $pCO_2$  rise, and not when the  $pCO_2$  decrease.

• **Response**: The sentence will be revised according to your comment.

In contrast to the long response times observed for discrete measurements at the 12 sites (Figure 4), the membrane-enclosed sensors reasonably performed well across most of river sections where  $pCO_2$  changed gradually (Figure 5). However, the measurements by the membrane-enclosed sensor exhibited substantial deviations from the measurements by the spray- and marble-type equilibrators where  $pCO_2$  changed abruptly during the period from12:20 to 12:35.

Line 430 - 432: What is the explanation to this drift? Not explained in text.

• **Response**: We will provide more explanations in the revised discussion.

During the field test, extraordinary algal blooms occurred as a combined result of a severe drought, warm temperatures, and high loads of nutrients discharged from water treatment facilities and polluted tributaries draining the Seoul metropolitan area. Chlorophyll-a concentrations increased from 21.1 mg m-3 on June 2nd to 46.7 mg m-3 on July 2nd (Water Information System of Korea; http://water.nier.go.kr). The "membrane sensor" might have been

more prone to biofouling of the sensor membrane by planktonic and associated bacterial communities than the other sensor protected by the Cu-mesh screen. Enhanced production or consumption of  $CO_2$  around the sensor membrane might have amplified diurnal fluctuations of  $pCO_2$ , leading to an increasing divergence between two sensor measurements as the time from maintenance increased.

Lines 432-434: "The duration during which relative differences of day-averaged  $pCO_2$  between the two sensors. . ."? I did not understand this section.

• **Response**: The sentence will be slightly revised in the revision.

Relative differences in daily means of  $pCO_2$  between the two sensors remained within 10 % during 5, 2, and 7 d since the routine maintenance on the 153th, 169th, and 182th day of the year, respectively.

Line 436: How you measure pH? What is the accuracy of the method? As you used the relationship pH- $pCO_2$  to examine the increasing biofouling effects with progressing time following the maintenance day, you must provide this information. Also, you need to show that the pH sensor not drift with time.

• **Response**: We used YSI 6820, as written in Line338. The accuracy of the pH probe was double checked on site with pH buffers and concurrent pH measurements using a portable pH meter (Orion 5-Star, Thermo Scientific, USA). This detail will be provided in the revised method section.

Line 438: How the biofouling can produced additional CO2 molecules? Explain in the text the process.

• **Response:** Explanation will be added in the revised text.

If additional  $CO_2$  molecules were produced by enhanced microbial activities in the biofilm formed over the surface of the sensor membrane, this could disturb the usual pH-CO2 relationship that can be explained by the carbonate equilibrium (Nimick et al., 2011).

Line 442-443: You pointed that "the method validation would require concomitant  $pCO_2$  measurements using other equilibration methods". You had all the possibilities to validate this method, but not did, i.e., you had large  $pCO_2$  variations and you had three equilibrator types.

• **Response:** We could not validate our sensor measurements with concurrent measurements using other methods. This limitation, together with further discussion of method validation, will be added in the revision.

Line 446: "Repeated maintenance visits at short intervals of 3-5 d may be required for a long term deployment of the sensor without antifouling measures in an inland water site with high levels and large diurnal fluctuations of  $pCO_2$ ." This is difficult depending of the study are. Do you have other suggestions?

• **Response:** That's the reason we recommended the antifouling practice. Short-interval maintenance would be very difficult to implement in many cases; therefore, antifouling measures should be prepared for long-term observation at eutrophic waters. Our suggestion was presented in Lines 450–460. The paragraph will be rewritten for clarity in the revision.

Line 460: Interesting result. Can you plot the graph showing these results for oligotrophic waters?

• **Response:** The results will be added as an additional figure or supplementary material in the

revised manuscript.

Table 1. Insert one column with the equilibration time for each method.

• **Response:** The change will be made in the revision.

For figures 2, 3, 4, 5, 6, 7, and 8 I think is better a white fill, without grades and with black contours.

• **Response**: The figures were drawn using default setting provided in ggplot2 of R. The default gray background theme was based on studies of visual impacts as stated by Wickham (2009; p. 141; original text provided below). Unless the gray background disturbs readability of the figures, we would like to respect the developer's intent. Nevertheless, we are open to revise the theme of the figures.

This (very light grey background with white gridlines) follows from the advice of Tufte (1990, 1997, 2001, 2006) and Brewer (1994a); Carr (1994, 2002); Carr and Sun (1999). We can still see the gridlines to aid in the judgement of position (Cleveland, 1993b), but they have little visual impact and we can easily "tune" them out. The grey background gives the plot a similar colour (in a typographical sense) to the remainder of the text, ensuring that the graphics fit in with the flow of a text without jumping out with a bright white background. Finally, the grey background creates a continuous field of colour which ensures that the plot is perceived as a single visual entity. (Wickham H. 2009. ggplot2: Elegant Graphics for Data Analysis. Springer, New York, p. 141)

---

## Author Response (AR1)

**Associate Editor**

Having now read your answers to the reviewer's comments and projected changes to the manuscript, I am happy to encourage you to proceed with the full revision of your manuscript. In addition to all the minor adjustments you have mentioned, I recommend you carefully address in your revised MS the following important points raised by both referees:

1. Shorten the "review" section, at least don't call it a review, just an extensive introduction

2. Provide all additional data required to satisfy the referee's criticisms

I am happy with the term "inland waters", although I suggest you add the term "flowing" (flowing inland waters) because most of your conclusions apply to lotic systems (Rivers / tidal rivers), but not to lentic environments. Alternatively use simply the term "river" in the title.

In addition, I suggest you remove Figure 9, which I found useless, and take benefit of the spare space to include more quantitative info (additional tests/data in text figures Tables) and qualitative info (discussion in text)

Looking forward to reading this soon.

- **Response:** We appreciate your constructive comments and suggestions. We have revised our manuscript incorporating all your comments and suggestions, as detailed below and in the subsequent responses to referee reviews. Please note that the revised manuscript has been checked again by a native English speaker to improve accuracy and readability.

   (1) We rewrote and shortened the Introduction and review sections. In the Introduction section we focused on the backgrounds and research needs for continuous $p$CO$_2$ measurements and then compared widely used gas equilibration systems in the second section (as a separate, extended introduction).

   (2) We included more data and descriptions on the three compared systems (Tables 1, S1; Figures 1, S1), laboratory response time tests (Figure 2), field comparisons of accuracy (Figure 3) and response time (Figure 4), underway measurements (Figure 5), and pictures (Figure S2) and the pH-$p$CO$_2$ relationship obtained in an oligotrophic reservoir (Figure S3).

   (3) In our view, the term "inland waters" best represents the study site, which includes a dammed middle reach and a downstream estuarine reach. Data from these diverse environments have been included in Figures 3, 4, 5 and S3.

   (4) We removed Figure 9 and provided additional descriptions and discussions through the manuscript. All changed texts are indicated by a blue color.

   Thank you very much for your consideration of our revised manuscript for publication in Biogeosciences!

**Referee #1**

**Major comments**

This paper provides an analysis of 3 commonly used equilibration systems for measurement of water column $pCO_2$ – the spray type "Weiss" equilibrator, the marble equilibrator and a membrane enclosed system. The authors present data from a series of laboratory and field experiments to assess the pros and cons of each system.

The paper claims to be a combine "literature review" and experimental paper, yet the review of the literature is somewhat limited. The title states it relates to "inland waters" but perhaps this is better changed to "freshwater systems" as there are little data from estuarine systems, which have been investigated thoroughly using the techniques described here. Equilibration systems have been reviewed rather extensively in the past, and the performance of the individual systems assessed here have already been detailed. The paper does present some new information on biofouling with the membranes systems which would be of interest to those using similar techniques.

The paper while generally well-written does require some editing to improve the readability/English (e.g. line15, line 56 what is high leverage of organic acids?, line 136 . . .which is mostly an IRGA. . . etc.).

- **Response overview:** We appreciate your constructive comments and suggestions. We have revised our manuscript incorporating all your comments and suggestions. The revised manuscript was checked again by a native English speaker to improve accuracy and readability. Some of your and another reviewer's major comments are overlapped, so the same overview of our common responses is provided below.

  **(1) Review**: There was a common critique on the novelty of our literature review; the review was evaluated as "somewhat limited" or "not a novelty". Therefore, the review section, along with the introduction, has been shortened and restructured. we focused on the backgrounds and needs for continuous $pCO_2$ measurements in the introduction section and compared widely used gas equilibration systems in the second section (as a separate, extended introduction).

  **(2) Additional monitoring data**: In response to the comments on the lack of measurements by the marble-type equilibrator in comparing the performance of the three equilibration systems, we conducted additional field measurements that would be useful when comparing the performance (e.g., accuracy and response time) of the three systems (Figures 3, 4).

  **(3) Methodological details**: More detailed descriptions of the compared gas equilibration systems, together with other in-situ measurements, such as pH, analytical procedures, and QC procedures, have been added in the Methods section, Tables 1 and S1, and Figures 1 and S1.

  **(4) Target water systems**: We used inland waters in the title because we also considered estuarine waters in literature review and our field study. Our study site includes a tidal reach of the Han River and data are included in Figures 3, 4, 5. Please also see the editor suggestion and our response to his suggestion.

**Specific comments**

Introduction - $CH_4$ is mentioned at line 30, but nowhere else, I suggest removing this reference as it gives the reader the expectation there will be some discussion about this.

- **Response:** We have removed this sentence and focused on $CO_2$ in the revision.

Methods - Some more details in the methods would also be helpful. For example was temperature and pressure measured within the marble and spray-type equilibrators, if not were the equilibrators vented to the atmosphere, and how were temperature differences between the water column and the equilibrator dealt with.

- **Response:** The descriptions on the methodological details in equilibration methods have been improved

through the text and in Figures 1 and S1. Measurements of temperature and the pressure inside the equilibrators is provided in Lines 196–203.

*The temperature differences between the river water and the equilibrator outflow water were usually within 0.3°C. The differences in barometric pressure between the inside and outside of the equilibrator chamber were lower than 5 μatm when the chamber vent was closed. The vent was closed during all the measurements after preliminary laboratory tests had confirmed that the small increase in the barometric pressure would not affect the accuracy of the $pCO_2$ measurement. The small initial pressure build-up immediately after turning on the water pump was relieved during ventilation for a few seconds through a vent channel that was established by using a three-way cock on the air-flow circuit. In addition, the integral pressure compensation function of the IRGA (LI820) we used was able to reduce any potential risk of inaccurate $CO_2$ analysis being induced by pressure changes.*

Line 256 0, 500, 5000, 10000 ppm?

- **Response:** Yes. The change has been made in Lines 216–217.

Line 269 I do not think one test on response time is adequate to draw too many conclusions – some replication would add some strength to this analysis. Also what about a high to low concentration step – this could take a considerable time in the membrane system. Do the authors have any explanation for the noisy response time data from the marble equilibrator? Also while t 95 and t100 has been used in the past, the best way to assess equilibration time are the models presented by Johnson 1999 [Johnson, J. E. Evaluation of a seawater equilibrator for shipboard analysis of dissolved oceanic trace gases. Anal. Chim. Acta 395, 119-132 (1999)].

- **Response:** Additional lab tests were conducted in response to your comment on differences in response time between low-to-high and high-to-low equilibrations and noisy response data from the marble equilibrator. A new figure (Figure 2) is now presented together with its descriptions in the text. Unlike Johnson (1999), who used the exponential decay or e-folding curve fitting, we found that the ideal exponential decay curve did not represent the various response patterns observed in our field tests, particularly for the membrane-enclosed sensor. Therefore, we had to opt for $t_{95}$ as used in other studies and described how we had determined $t_{95}$ in Lines 229–234.

Line 303 – Can the authors give some details about how this 10 km/h speed was determined? It seems too fast to assess changes over a 10 km stretch of river (i.e. 1 hour transit time)

- **Response:** We determined the boat speed based on our own field tests and previous studies such as Crawford et al. (2015). The good agreements between underway measurements and spot manual headspace equilibration measurements indicate that an appropriate boat speed was determined. We have included more data over the entire survey period > 4 hr in Figure 5 and more detailed descriptions are provided in Lines 263–268.

Line 322 The reader is initially given the impression that the 3 systems will be compared for the studies – yet the 3 systems are only compared for the survey data. Perhaps this can be clarified earlier, or in the title

- **Response:** As described in the relevant parts of the original manuscript, logistic constraints forced us to select one or two equilibrators because multi-site tests were conducted as part of another monitoring program. However, we conducted more comparison tests including the marble-type equilibrator. New data on the performance of the marble-type equilibrator system have been included in Figures 2–4.

Line 344 Was the data corrected for equilibration time in the regression analysis?

- **Response:** Yes, all compared data were taken as pCO2 values after passing a specific equilibration time.

Line 350 Can the authors give a bit more detail about what the aim of this analysis is?

- **Response:** The following sentences have been added in Lines 311–314.

> *The analysis was based on the assumption that robust pH–pCO₂ relationships could be expected from the carbonate equilibrium model if there were no artifact effects such as sensor biofouling. The temporal changes in the pH–pCO₂ relationships were examined to assess the biofouling-induced deviations from the robust pH–pCO₂ relationship*

Line 380-382 This is also due to the difference in diffusivity between the water-air interface (spray and marble equilibrators) and the water PTFE interface.

- **Response:** We have added more discussion on this issue in Lines 348–352.

  *The diffusion-type IRGA of the membrane-enclosed sensor generally exhibited longer response times compared with those of the flow-through IRGAs of the equilibrator systems. The passive gas transfer to the sensor unit could contribute to the longer response time of the membrane-enclosed sensor. In addition, the gas diffusivity across the water–membrane interface could differ from the diffusivity between the water–air interfaces within the equilibrator chambers.*

Line 384 – 391 What about the effect of temperature on diffusivity?

- **Response:** Temperature could be one of factors that regulate response time. We did additional analysis on temperature effects, but we could not find any effect because temperature did not vary a lot between sampling sites. Following sentence has been added in Lines 363–365.

  *The temperature could also affect response time, although regression analysis did not indicate any significant relationship between the temperature and response time, probably because of the relatively narrow range of temperature variations among the sampling sites.*

Line 395 I would suspect that allowing only 1 x response time for point measurements would not allow for any changes in the ambient changes in $pCO_2$ during the measurement interval.

- **Response:** By adding some extra time to the mean response time determined during the field tests, we wanted to ensure an adequate deployment time required to cover changes in ambient $pCO_2$ during a spot measurement. To clarify, the sentences have been revised in Lines 369–370.

Line 401 – To me it looks like the marble equilibrator gives consistently higher $pCO_2$ values for the elevated $pCO_2$ areas of the river. Do the authors have an explanation for this? Was pressure measured in the equilibrators? Was temperature measured in the equilibrators? These are very important measurements to make!

- **Response**: Some deviations of the marble-type equilibrator from other systems were observed only during the periods of sudden fluctuation of $pCO_2$. Potential causes for observed differences have been described in Lines 383–384. Please also note that we have added more data in Figure 5, which show general correspondence between the different systems. As described in a previous response, differences in water temperature and pressure were too small to affect $pCO_2$ measurement accuracy.

Line 408 - Do the authors mean "stationary" rather than discrete measurements (discrete implies headspace measurements)

- **Response:** A new term "spot measurement" has been consistently used through the manuscript to refer to spot measurements of $pCO_2$ in comparison to continuous underway or long-term measurements.

Line 419 – This has been done in estuaries in recent times, again perhaps expand to include estuaries in the analysis or use more specific terminology rather than inland waters

- **Response:** Please refer to our response to the same terminology issue.

Line 438-439 Biofouling could cause a shift either way ($CO_2$ increase or decrease) depending upon the community composition.

- **Response:** In response to your comment, the sentence has been revised in Lines 424–426.

  *If additional $CO_2$ molecules were produced or consumed by the biofilms formed on the membrane sensor, it could disturb the usual pH–$pCO_2$ relationship, which could be explained by the carbonate equilibrium model*

Figure 3 – I would recommend not using a log scale as this hides some of the differences between equilibrators. Alternatively if the authors add the measured values to the figure (perhaps at 90 degree angle within each bar) that would allow the reader to easily see how the systems compare

- **Response:** A new version (Figure 3) has been created without using a log scale to allow readers to directly compare the original measurement results of the three compared systems.

**Reviewer #2**

**General Comments**

The paper under review for biogeosciences presents a literature review combined with laboratory and field tests to evaluate the application potential of three widely used automated equilibration systems to continuous long-term or underway $pCO_2$ measurements. The paper is generally well-written and easy to follow, but I found some grammar and sentence structure issues and I am not a native speaker. I also found some inconsistences between the figure and the results descriptions (see specific comments). In addition, some objectives of the study were not achieved.

As you wrote "This study aims to review advantages and disadvantages of widely used $pCO_2$ equilibration methods and automated equilibration systems that can be used for continuous monitoring of highly variable $pCO_2$ across". The ''short review" of this paper with the advantages and disadvantages of widely used $pCO_2$ equilibrators is not a novelty for continuous aquatic $pCO_2$ measurements. The studies of Santos et al. 2012 and Webb et al. 2016 (including others) presents laboratory step experiments on six different equilibrators to constrain $CO_2$ equilibration time constants and short reviews of the equilibration technique, including shower-head, marble and membrane type equilibrators.

I think you must focus on the new information that this paper provide about improvements in the aquatic $pCO_2$ measurements, which are the long-term deployment of the equilibrators under various field conditions and biofouling with the membrane systems. You must to describe the equilibrator systems with details (the systems were poorly described).The figure 1 and the text did not present details of the measurement-systems, and I think this is very important. In addition, some tests were not performed for the marble equilibrator. I think that is important provide one or two tables with the field and laboratory test results.

You compare the drifts of the $pCO_2$ results for the membrane equilibrators comparing the relationship between pH and $pCO_2$ during successive 4-day monitoring periods following maintenance. However, pH measurement method is missing in the Method section. Since it is used to evaluate the reliability of measured $pCO_2$, it must be evaluated more rigorous. Apparently, the problematic of long-term monitoring of $pCO_2$ is still unsolved (the drifts of the results are very high if is not applied continuous maintenance of the measuring system).

- **Response overview:** We appreciate your constructive comments and suggestions. We have revised our manuscript incorporating all your comments and suggestions. The revised manuscript was checked again by a native English speaker to improve accuracy and readability. Some of your and another reviewer's major comments are overlapped, so the same overview of our common responses is provided below.

  **(1) Review**: There was a common critique on the novelty of our literature review; the review was evaluated as "somewhat limited" or "not a novelty". Therefore, the review section, along with the introduction, has been shortened and restructured. we focused on the backgrounds and needs for continuous $pCO_2$ measurements in the introduction section and compared widely used gas equilibration

systems in the second section (as a separate, extended introduction).

**(2) Additional monitoring data**: In response to the comments on the lack of measurements by the marble-type equilibrator in comparing the performance of the three equilibration systems, we conducted additional field measurements that would be useful when comparing the performance (e.g., accuracy and response time) of the three systems (Figures 3, 4).

**(3) Methodological details**: More detailed descriptions of the compared gas equilibration systems, together with other in-situ measurements, such as pH, analytical procedures, and QC procedures, have been added in the Methods section, Tables 1 and S1, and Figures 1 and S1.

**(4) Target water systems**: We used inland waters in the title because we also considered estuarine waters in literature review and our field study. Our study site includes a tidal reach of the Han River and data are included in Figures 3, 4, 5. Please also see the editor suggestion and our response to his suggestion.

**Specific Comments**

Line 11: Replace for emissions.

- **Response:** The "evasion" has been replaced by "emission" in Line 10.

Line 18: '. . .upper detection limit of the sensor". What is this limit?

- **Response:** We also think that this sentence would be confusing without more detailed descriptions, so it has been rewritten in Line 17.

  *...along the river sections where $pCO_2$ varied within the sensor detection range.*

Line 17: The overall results suggest that the equilibrators are better suited for relatively short underway measurements than long-term deployment. Why? Do you have suggestions to improve the equilibration systems in order to long-term $pCO_2$ monitoring? I think you must discuss better this point.

- **Response:** Our suggestions have been added in the newly written conclusions (Lines 463–484). The sentence in the abstract has been rewritten in Lines 18–24.

  *The overall results suggest that the fast response of the equilibrator systems facilitates capturing large spatial variations in $pCO_2$ during relatively short underway measurements. However, the attendant technical challenges of these systems, such as clogging and desiccant maintenance, have to be addressed carefully to enable their long-term deployment. The membrane-enclosed sensor would be suitable as an alternative tool for long-term continuous measurements, if membrane biofouling could be overcome by appropriate anti-fouling measures such as copper-mesh coverings.*

Line 26: First sentence confuse.

- **Response:** The sentence has been rewritten in Lines 26–27.

  *Recent synthesis efforts have highlighted the importance of carbon dioxide ($CO_2$) emissions from inland waters in the global carbon cycle.*

Line 27: I think the "respiration" is more adequate.
Line 26: I think "emission" or "degassing" is better than evasion.

- **Response:** The specific sentence was removed while shortening the Introduction section. "Emission" has been consistently used throughout the manuscript.

Please review all the references. I found some mistakes.

- **Response:** We have thoroughly checked any mistakes in the cited references.

Line 30: You wrote $CO_2$ and $CH_4$. I think is better write dioxide carbon ($CO_2$) and methane ($CH_4$).

- **Response:** The changes have been made in Line 26 and 95.

Line 35 and 36: Confused. You must explain better the principles of direct and indirect measurements. This sentence is not clear and not sufficient.
Line 36: ". . . between water and air and gas transfer. . ."? This is not clear, please rewrite.

- **Response:** To clarify, the sentence has been rewritten in Lines 44–45.

  *The $CO_2$ emission rate can be determined either by directly measuring the transfer of $CO_2$ across the water–air interface, or by estimating the flux based on (1) differences in the $pCO_2$ between the water and air, and (2) the gas-transfer velocity.*

Line 39-44: Please read and include information of Lorke et al. (2015) paper. There are important considerations about the floating chamber measurements and improvements on this technique to application for running waters.

- **Response:** Thank you for the useful reference. The paper has been cited in Lines 47–53.

  *However, the attendant technical challenges include the difficulty of deploying the floating chamber stably over often turbulent water surfaces and the disrupted natural turbulence inside the floating chamber that could result in overestimations of the $CO_2$ flux (Vachon et al., 2010), especially when the chamber is anchored at a fixed spot (Lorke et al., 2015). Recently, Lorke et al. (2015) have proposed improved designs for floating chambers that minimize the bias of the gas-transfer velocity, including a freely drifting chamber on running water, or an anchored chamber with a close contact over the water surface. However, validation and further technical improvements are needed before these proposed chamber systems could be applied in practice.*

Line 50: from $pCO_2$ measurements.

- **Response:** The change has been made in Line 59.

Line 51 and 52: You can also calculated $pCO_2$ from dissolved inorganic carbon (DIC) and total alkalinity (TA) and ancillary parameters. Please include this information. You can read Dickson (2010) to include more accurate statements about the indirect calculations of $pCO_2$.

- **Response:** The change has been made in Lines 61–62.

  *In addition, the $pCO_2$ can be estimated from two of the three variables in the carbonate equilibrium model, namely, pH, alkalinity, and dissolved inorganic C (DIC) (Lewis et al., 1998; Dickson et al., 2007).*

Lines 61 and 62: SOCAT?

- **Response:** The "Surface Ocean $CO_2$" has been added in Line 30.

Line 70: delete "from polluted waterways" ?

- **Response:** The sentence has been rewritten in Line 40.

  *the evasion of $CO_2$ from "polluted waterways" in urbanized watersheds → the emission of $CO_2$ from urbanized inland waters*

You did not present the results of the tests (you must insert one or two tables with the results of the field and

laboratory tests).

- **Response:** It is not clear to which parts the reviewer refer. We hope that the revised manuscript has contained all necessary data to address the reviewer's concerns.

Figure 1: This information is not sufficient. Please, provide more details about the measurement systems of $pCO_2$. For example, see the Figure 1 in Frankignoulle et al. (2001). As you stated that "Our review and cross-validation tests focus on three automated equilibration systems: spray- and marble-type equilibrators and a membrane-enclosed sensor (Table 1)" you must provide more details about the functioning and details of these three systems.

- **Response:** More detailed descriptions have been added in the Methods section (Lines 173–224) and in the revised Figure 1 and Table 1, also including the photos of three systems in Figure S1 and the lists of components of the equilibration systems in Table S1.

Line 121 – 124: Bakker et al. (1996) measuring $pCO_2$ in estuarine waters, found "Frequent blockage of the showerhead of the equilibrator with algal material", adding some problems to the measurements. I would like to see some discussion about this problem with the equilibrators.

- **Response:** The paper was cited to discuss potential blockage of the equilibrators in Lines 394–396:

  *Bakker et al. (1996) reported on frequent blockages of their showerhead equilibrator with particulate materials derived from algal blooms in Dutch coastal waters. Long deployments of the spray-type equilibrator in eutrophic freshwaters could also result in similar clogging problems.*

  A further discussion has been included in lines 471–472:

  *However, further tests are required to determine how long the marbles and the nozzle could remain unaffected by biofouling or clogging during continuous deployment over several hours to days.*

Lines 129 – 138: A figure with more details of the systems can better elucidate this section of the paper.

- **Response:** Specific features of equilibration systems are now presented in more detail in the revised Figure 1 and Table 1 and the new Figure S1, with further descriptions provided in the supplementary information (Table S1).

Lines 150-152: Do you have some suggestions to turn the equilibrations systems (marble type and showerhead) more automated for long-term monitoring? Please discuss possible improvements that are necessary for long-term monitoring.

- **Response:** Some maintenance and technical recommendations have been provided in the Conclusions section (Lines 472–475).

  *To address potential clogging and blockage problems of the equilibrators, spare sets of the equilibrator chamber should be prepared during underway measurements. An automated switching between replicate equilibrator chambers at pre-fixed intervals could help to extend the monitoring duration.*

Line 154-156: Again, here add one figure can better illustrate how is the passive membrane $CO_2$ equilibration systems, providing details for easy reproducibility.

- **Response:** Same as the previous response.

Line 159 – 161: "There are a small number of commercially available membrane enclosed sensor systems (e.g., eosGP, Eosense Inc., Canada; Mini-Pro CO2, Pro-Oceanus Systems Inc., Canada)". What are the lower and upper detection limits of these sensors? They can be applied in aquatic systems where the $pCO_2$ values can easily be higher than 10,000 ppmv?

- **Response:** The detection ranges can be adjusted by manufacturer, from low range (e.g., 0–1000 ppm) to high range (e.g., 0–20%). The sensors are able to cover the range of $pCO_2$ in inland waters when the detection ranges are appropriately set and calibrated. This wide detection range has been mentioned in Lines 130–131.

  *Some of the $CO_2$ sensors used in these commercial systems can detect a wide range of $CO_2$, covering the usual range of $pCO_2$ found in inland waters.*

Lines 220-222: A range of 2000 ppmv is "high" in inland waters. Then, this type of system (membrane-enclosed senor) could not be used in some environmental conditions. In addition, I would like to see some discussions about these overestimations.

- **Response:** The cited paper did not elaborate on potential sources of overestimation in the high $pCO_2$ range, so it would not be possible and appropriate to provide explanations based on our visual inspection. Therefore, we deleted the sentence.

Lines 233-235: This information is not sufficient. You must provide details of the instruments.

- **Response:** Aforementioned more detailed information about the systems have been included in the revised manuscript (Tables 1, S1; Figures 1, S1).

Line 242 – 245. Are you sure that the unit is "mm"? One acrylic tube with this measure is very small, and I think cannot be filled with glass marbles. For example, in Frankignoulle et al. (2001) the vertical Plexiglas measures were: height 80 cm; diameter 10 cm.

- **Response:** Yes. We devised a smaller system by modifying it from the original marble-type equilibrator of Frankignoulle et al. (2001). The smaller chamber was filled with smaller marbles (diameter: 10 mm) than the larger 20–30 mm marbles used for the bigger chamber. Other researchers have also used small systems for portability; for instance, Abril et al., (2006) used a chamber of 8 cm in diameter and 60 cm in height. We compared small-sized and original equilibrators in laboratory and we found no significant differences in performance between them. We have noted this information in Lines 186–188.

  *A marble-type equilibrator, smaller than those used in previous studies (Frankignoulle et al., 2001; Abril et al., 2006), was designed, based on laboratory tests to enhance the portability of the device without compromising the measurement accuracy.*

Line 250-255: Provide a detailed picture of the complete system.

- **Response:** As mentioned before, the change has been made in the revision.

Line 252: Despite the fact that Johnson et al. (2010) provided details of the membrane enclosed sensor, this is not sufficient for publish in biogeosciences. Your work must yield descriptions of the equilibration systems, both in text and in figures. Your third objective was "to compare the accuracy and maintenance requirements of three selected equilibration systems (a spray- and a marble-type equilibrators and a membraneenclosed $CO_2$ sensor) for field applications in a series of laboratory and field crossvalidation tests". I think that your objective is not just this, rather, I think that is also describe with details these three selected equilibration systems.

- **Response:** Again, we have added more information about the system in text and figure in the revision.

Line 256-257: "The $CO_2$ analyzers and sensors were calibrated in the laboratory using $CO_2$ gases of known concentrations (0, 500, 500, and 10,000 ppm) immediately before each laboratory or field test." Why two concentrations of 500? Did you make the calibration after the field test to see the drift of the sensors?

- **Response:** The mistake of the wrong $CO_2$ gas concentration has been corrected (500 → 5,000 ppm). Yes, the sensors were also checked following deployment and we found little drift during the deployment periods from which the presented results were obtained, as described in Lines 216–219.

Line 262: About the laboratory test, why you did make the first test just to the membrane-enclosed sensor? This test is not well explained, please rewrite.

- **Response:** Another version is now used to present new test results (Figure 2).

Line 275: you did not perform the field test for the marble-type equilibrator. One of your objectives was not realized, since you compared the field tests just for 2 equilibrators. Despite the fact that you assumed that both types would exhibit similar results based on the laboratory results, this cannot be true in field conditions.

- **Response:** We have included additional measurements using the marble-type equilibrator in Figures 2–4 . We hope that new data provide a more thorough comparison of the three systems.

Line 295: "The response time was determined as the full time ($t100$) or 95% of the full time ($t95$) it took to a final $pCO_2$ level that represents $pCO_2$ values exhibiting less than 1 % of coefficient of variation (CV) for 2 min." The full time ($t100$) is unusual for calculations of equilibration time.

- **Response:** Only $t_{95}$ values have been used in the revised manuscript. The descriptions on response time calculation have been rewritten in Lines 229–234.

  *The response time ($t_{95}$) was determined as the time required to reach the 95 % level of the final stabilized $pCO_2$ values that exhibited variations smaller than 1 % of the coefficients of variation (CV) for 2 min. In addition, the response time can be assessed by calculating the time constant ($\tau$) of the exponential or e-folding curve fitting of varying $pCO_2$ values during high-to-low equilibration (Johnson, 1999). As the various response patterns observed for the membrane-enclosed sensor could not be fitted by the ideal exponential decay curve, we present only $t_{95}$ results.*

Line 303: How were the prior tests of boat speed effect?

- **Response**: As already mentioned in an earlier response, we determined the boat speed based on our own field tests and Crawford et al. (2015). The good agreements between underway measurements and spot manual headspace equilibration measurements indicate that the boat speed was not too fast. More detailed descriptions are provided in Lines 265–268.

Line 314: I think the upper detection limit of the membrane sensor are low and cannot be applied in several inland waters where the natural variations of $pCO_2$ are very higher than these limits.

- **Response**: The detection ranges can be adjusted by manufacturer, various detection ranges, from low ranges (e.g., 0–1000 ppm) to high ranges (e.g., 0–20%). The sensor we used and other commercially available sensors can cover the wide range of $pCO_2$ in inland waters, as described in Lines 130–131.

Line 320-322: This section is confuse. Please rewrite. What preliminary tests did you perform? Why the power supply and air flow dehydration were easiest for the spraytype equilibrator?

- **Response**: The sentences has been rewritten as follows: In addition, in our experience, the marble-type equilibrator consumed much desiccant than the spray-type equilibrator, probably due to differences in the ratio of headspace air and water. Moreover, we worried about an accidental flooding to air circuit by flowing backward of the marble-type equilibrator during an unmanned field deployment.

  *The spray-type equilibrator was selected for use, as the preliminary tests had shown that it was easier to maintain the power supply and air-flow dehydration with this type of equilibrator than with the marble-type.*

Line 324: Why you did not test the marble-type equilibrator for the long-term measurements?

- **Response**: Our tests with both equilibrators were not successful for the same reason as described in Lines 281–283.

Line 338-342: Move to results section.

- **Response**: The change has been made in the revision (Lines 414–418).

Line 345: Did you test the normality of data set? If not follow a normal distribution, you cannot apply the t-test. You must apply the non-parametric tests as Wilcoxon, for example.

- **Response**: Yes, we checked the normal distribution using Shapiro-Wilk test, as described in Lines 304–305.

Line 349: What are the pH-$pCO_2$ relationships? Not clear in the text. Since it is used to evaluate the reliability of measured $pCO_2$, it must be evaluated more rigorous.

- **Response:** Following sentences have been added in Lines 311–314.

  *The analysis was based on the assumption that robust pH–pCO₂ relationships could be expected from the carbonate equilibrium model if there were no artifact effects such as sensor biofouling. The temporal changes in the pH–pCO₂ relationships were examined to assess the biofouling-induced deviations from the robust pH–pCO₂ relationship.*

Lines 363 – 365: The results in figure 2 shows that for low $pCO_2$ values the coefficient of variation calculated from the compared measurements were higher than 10%. However, in the text you not explain why this occurs for low $pCO_2$ values. Line 364: "The CV values were smaller than 5% at all sites except site 3 and 8. . .". The graph did not show this. Sites 1, 2, 4, 7, 10, 11, 12 CV < 10%. Sites 5, 6, 9 CV < 5%. Sites 3 and 8 CV > 10%. For only 3 sites the CV values were smaller than 5 %, please correct.

- **Response**: A new version (Figure 3) presents more data comparing all the three equilibration systems.

Lines 373-374: "The response time of the spray-type equilibrator falls within the usual range of response times reported for the spray- (8 min; Santos et al., 2012). . ." Not really. You tests were approximately 4 time more rapid than that reported by Santos et al., 2012 for the spray-type equilibration. You should point some suggestions to explain this difference. Why you did not perform the equilibration time test for the marble-type equilibrator? Where are the results?

- Additional field data using the marble-type equilibrator have been included in the revision, so a new description is provided in Lines 335–343.

  *The mean t₉₅ was 1 min 45 s for the spray-type equilibrator and 2 min 5 s for the marble-type equilibrator, without showing noticeable differences between the standing and flowing waters. The response time of the spray-type equilibrator was shorter than the response times reported by another study (8 min; Santos et al., 2012), but were similar to the response times of the marble-type equilibrators reported by other studies (2–3 min; Frankignoulle et al., 2001; Abril et al., 2014). The differences in the response time of the spray-type equilibrators could be ascribed to various factors, including the different levels of pCO₂ (~100–10,000 μatm in this study vs. > 10,000 μatm in the study by Santos et al., 2012), the equilibrator size (251 vs. 1963 cm⁻³), the length of the air circuit, and the performance of the spray nozzle.*

Line 384: The figure 4 not showed a logarithm curve, rather, showed a linear tendency.

- **Response**: The curve appears linear because of the log-scaled X axis. To clarify, "note a log scale for the x-axis" has been added in the caption of Figure 4.

Line 385: ". . .with steeper increases observed for the membrane-enclosed sensor, particularly in flowing waters." Is not the contrary? The steeper increase seems to be to the standing waters (Figure 4; red circles)."

- **Response**: Yes, the increase was steeper in the standing waters. The sentence has been corrected in Line 357.

Line 394: and expected range of $p\text{CO}_2$ levels.

- **Response**: The change has been made in Line 367.

Line 405–407: "although it failed to respond to rapid $p\text{CO}_2$ increases from the relatively low value at the confluence (11:57) to the concentration peak (12:25) due to the limited detection range." Not just this. If you look at 12:15 and at 12:30, the deviation seems substantial also when the $p\text{CO}_2$ values decreased abruptly. You just discussed the deviation when the $p\text{CO}_2$ rise, and not when the $p\text{CO}_2$ decrease.

- **Response**: The sentence has been revised in Lines 383–386.

  *The sensor measurements also deviated noticeably from the measurements obtained with the spray- and marble-type equilibrators for the period from 12:20 to 12:35, during which the pCO₂ changed abruptly. In contrast with the long response times observed for spot measurements at the 26 sites (Fig. 4), the membrane-enclosed sensors exhibited good agreements with the other results across most of the river sections where pCO₂ changed relatively gradually (Fig. 5).*

Line 430 - 432: What is the explanation to this drift? Not explained in text.

- **Response**: We have provided more explanations in the revised discussion (Lines 414–421).

  *During the monitoring period, extraordinary algal blooms occurred that were ascribed to a combination of factors, such as severe drought, warm temperatures, and high loads of nutrients discharged from water treatment facilities and the polluted tributaries draining the Seoul metropolitan area. The chlorophyll-a concentration increased from 21.1 mg m⁻³ on 2 June to 46.7 mg m⁻³ on 2 July (Water Information System of Korea; http://water.nier.go.kr). The bulk membrane sensor could have been more prone to biofouling by planktonic and associated bacterial communities than the membrane+Cu sensor was. Enhanced production or consumption of CO₂ around the sensor membrane could have amplified the diurnal fluctuations of pCO₂, leading to considerable divergence between the two sensor measurements with increasing time after maintenance.*

Lines 432-434: "The duration during which relative differences of day-averaged $p\text{CO}_2$ between the two sensors. . ."? I did not understand this section.

- **Response**: The sentence has been rewritten in Lines 413–414.

  *The relative differences in the daily mean pCO₂ between the two sensors remained within 10 % for 5, 2, and 7 d after the routine maintenance on the 153th, 169th, and 182th day of the year, respectively.*

Line 436: How you measure pH? What is the accuracy of the method? As you used the relationship pH-$p\text{CO}_2$ to examine the increasing biofouling effects with progressing time following the maintenance day, you must provide this information. Also, you need to show that the pH sensor not drift with time.

- **Response**: We used YSI 6820, as written in Line 299. The accuracy of the pH probe was regularly double checked on site with pH buffers and concurrent pH measurements using a portable pH meter (Orion 5-Star, Thermo Scientific, USA). These details have been provided in Lines 299–301.

Line 438: How the biofouling can produced additional $\text{CO}_2$ molecules? Explain in the text the process.

- **Response:** Potential causes are provided in Lines 424–426.

  *If additional CO₂ molecules were produced or consumed by the biofilms formed on the membrane sensor, it could disturb the usual pH–pCO₂ relationship, which could be explained by the carbonate equilibrium model (Nimick et al., 2011).*

Line 442-443: You pointed that "the method validation would require concomitant $p\text{CO}_2$ measurements using other

equilibration methods". You had all the possibilities to validate this method, but not did, i.e., you had large $p$CO$_2$ variations and you had three equilibrator types.

- **Response:** We could not validate our sensor measurements with concurrent measurements using other methods, primarily because large increases in $p$CO$_2$ usually occurred at night whereas our maintenance visits were conducted during day. We mentioned the need of further verification in the Lines 429–430.

Line 446: "Repeated maintenance visits at short intervals of 3 – 5 d may be required for a long term deployment of the sensor without antifouling measures in an inland water site with high levels and large diurnal fluctuations of $p$CO$_2$." This is difficult depending of the study are. Do you have other suggestions?

- **Response:** That's the reason we recommended the antifouling practice. Short-interval maintenance would be very difficult to implement in many cases; therefore, antifouling measures should be prepared for long-term observation in eutrophic waters. Our suggestion was presented in Lines 482–486.

Line 460: Interesting result. Can you plot the graph showing these results for oligotrophic waters?

- **Response:** The results are provided as supplementary information (Figures S2, S3).

Table 1. Insert one column with the equilibration time for each method.

- **Response:** The change has been made in the revision.

For figures 2, 3, 4, 5, 6, 7, and 8 I think is better a white fill, without grades and with black contours.

- **Response**: The figures were drawn using default setting provided in ggplot2 of R. The default gray background theme was based on studies of visual impacts as stated by Wickham (2009; p. 141; original text provided below). Unless the gray background disturbs readability of the figures, we would like to respect the developer's intent. Nevertheless, we are open to revise the theme of the figures.

  *This (very light grey background with white gridlines) follows from the advice of Tufte (1990, 1997, 2001, 2006) and Brewer (1994a); Carr (1994, 2002); Carr and Sun (1999). We can still see the gridlines to aid in the judgement of position (Cleveland, 1993b), but they have little visual impact and we can easily "tune" them out. The grey background gives the plot a similar colour (in a typographical sense) to the remainder of the text, ensuring that the graphics fit in with the flow of a text without jumping out with a bright white background. Finally, the grey background creates a continuous field of colour which ensures that the plot is perceived as a single visual entity*. (Wickham H. 2009. ggplot2: Elegant Graphics for Data Analysis. Springer, New York, p. 141)

[revised manuscript text omitted]

715    **Figure 3.**

[Figure]

**Figure 4.**

[Figure]

720

**Figure 5.**

[Figure]

**Figure 6.**

[Figure]

**Figure 7.**

[Figure]

730    **Figure 8.**

---

## Author Response (AR2)

**Associate Editor**

Dear authors
Having now read the revised version of your technical note, I conclude it should be accept for publication in BG, after some minor revision. I made my own review of your MS and I suggested some changes on several points that need clarification before publication. Looking forward reading a revised version of your paper.
With best regards, Gwenaël Abril

- **Response:** We appreciate your positive comments and valuable suggestions. We have revised our manuscript incorporating all your comments and suggestions, as detailed below. Thank you very much for your time and effort!

L55 eddy covariance measurements also include the paper by Polsenaere et al (2013)
Polsenaere P., Deborde J., Detandt G., Vidal L.O., Pérez M.A.P. and Abril G. (2013) Thermal enhancement of gas transfer velocity of CO2 in an Amazon floodplain lake revealed by Eddy Covariance. Geophysical Research Letters 40, 1734–1740, doi:10.1002/grl.50291.

- **Response:** The paper has been cited in Line 55.

L191 what was the motivation for choosing a different water flow in each equilibrator type? Later in the discussion, water flow must be considered as an important parameter that affects response time.

- **Response:** Different optimal water flow rates were determined in preceding laboratory tests to enable fast equilibration without causing chamber overflow, as newly described in Lines 190–192. The potential effect on response time has also been mentioned in Lines - .

L217 please specify which $CO_2$ concentration was selected to span the LICOR, and which ones were used for linearity check

- **Response:** More descriptions of the $CO_2$ analyzer QC are provided in Lines 219–221.

*The IRGA used for the two equilibrators (LI820) was spanned with a pure N2 gas (>99.9%) and a CO2 standard gas at 10,000 ppm. Linearity check was performed with two additional CO2 standard gases (500 and 5,000 ppm).*

L328 change "good agreement" to "excellent agreement" (R>0.99)

- **Response:** The change has been made in Line 331.

L335-336 please consider that faster equilibration in the spray-type compared to the marble-type can be due to differences in water flow (2.5 L min-1 versus 1.5 L min-1)

- **Response:** Following your suggestion, a new sentence has been added in Lines 340–341.

*The small difference in response time between the equilibrators might have resulted from different operational conditions including the difference in water flow (2.5 and 1.5 L min-1 for the spray- and marble-type equilibrator, respectively).*

L345 to L353, avoid repetition on the fact that passive equilibration is slower

- **Response:** The redundant sentences have been rewritten to avoid repetition in Lines 351-354.

*The longer response time of the membrane-enclosed sensor can be explained by the fact that passive equilibration occurs without any physical process to facilitate equilibration underwater (Santos et al., 2012). Moreover, the diffusion-type IRGA of the membrane-enclosed sensor generally exhibited longer response times compared with those of the flow-through IRGAs of the equilibrator systems.*

L378 Frankignoulle et al. have compared equilibrator data with $p$CO$_2$ calculated from pH and alkalinity, and not headspace equilibration.

- **Response:** The citation has been replaced by Abril et al. (2006) in Line 382.

L381 what does "(approximately 12:00) refer to ? please clarify

- **Response:** It has been clarified as follows (Lines 384-386).

*However, after 12 o'clock the measurements deviated substantially as the boat entered the river segments where the inflow from a highly polluted tributary enriched in pCO2 elevated the pCO2 of the main stem above the upper detection limits of the two different sensors (~10,000 and 7,000 ppm)*

L382 provide the value of upper detection limits of the sensors.

- **Response:** The change has been made in Line 386.

L420 change "could have amplified" by "apparently amplified"

- **Response:** The change has been made in Line 424.

L437 and throughout the MS: change "spot" to "discrete"

- **Response:** The term "spot" was replaced by "discrete" throughout the MS.

L446 simplify: "Other mechanical antifouling techniques (e.g., brushing and wiping) could be applied to the membrane-enclosed sensor system; nevertheless, the copper-mesh screen could be superior for long-term pCO2 monitoring programs that require easy deployment, minimal maintenance, and low energy demand. " to something like "Other mechanical antifouling techniques (e.g., brushing and wiping) could be applied to the membrane-enclosed sensor system, but would consume more energy"

- **Response:** The change has been made in Lines 450-451.

L468 & L492 "spot measurements/sampling" > "discrete measurements/sampling"

- **Response:** The change has been made.

L498 remove the last sentence "Our technical recommendations and caveats can form a solid empirical basis for further studies that is required to improve the performance and maintenance of gas equilibration systems during continuous $p$CO$_2$ monitoring in a wide range of inland waters." Let's let the scientific community involved in $p$CO$_2$ measurements decide for that in the future.

- **Response:** The change has been made.

[revised manuscript text omitted]

725    **Figure 5.**

[Figure]

**Figure 6.**

[Figure]

730

**Figure 7.**

[Figure]

**Figure 8.**